# Deep learning image segmentation reveals patterns of UV reflectance evolution in passerine birds

Yichen He [1] ✉, Zoë K. Varley[1], Lara O. Nouri [1], Christopher J. A. Moody[1], Michael D. Jardine [1], Steve Maddock [2], Gavin H. Thomas [1,3] ✉ & Christopher R. Cooney [1] ✉

Ultraviolet colouration is thought to be an important form of signalling in many bird species, yet broad insights regarding the prevalence of ultraviolet plumage colouration and the factors promoting its evolution are currently lacking. In this paper, we develop a image segmentation pipeline based on deep learning that considerably outperforms classical (i.e. non deep learning) segmentation methods, and use this to extract accurate information on whole-body plumage colouration from photographs of >24,000 museum specimens covering >4500 species of passerine birds. Our results demonstrate that ultraviolet reflectance, particularly as a component of other colours, is widespread across the passerine radiation but is strongly phylogenetically conserved. We also find clear evidence in support of the role of light environment in promoting the evolution of ultraviolet plumage colouration, and a weak trend towards higher ultraviolet plumage reflectance among bird species with ultraviolet rather than violet-sensitive visual systems. Overall, our study provides important broad-scale insight into an enigmatic component of avian colouration, as well as demonstrating that deep learning has considerable promise for allowing new data to be brought to bear on long-standing questions in ecology and evolution.

The diversity of animal colouration is among the most striking features of life on Earth. This diversity arises through selection pressures relating to, for example, signalling (social and sexual), camouflage and crypsis, thermoregulation, and parasite defence[1,2]. The role of colouration in signalling is particularly complex because effective visual communication depends on both the strength of signal and perception of the receiver[3]. Selection is expected to strongly favour adaptations that maximise perception of the signal relative to background noise in the signalling environment[4]. Fundamentally, visual communication therefore depends on the visual sensitivity of the receiver and on the light environment. The light environment itself is determined by the available light spectrum resulting from filtered solar irradiation. For example, woodland and forest canopy habitats are dominated by ambient light rich in blue and UV[5].

In birds, visual signalling is a dominant mode of communication and diurnal birds in particular are highly sensitive to colour. However, not all birds perceive colour equally. Visual systems in birds can be classified as either violet sensitive (VS, with cone peak sensitivity from 402–426 nm and 50% of the incident light on the cornea transmitted to the retina as low as -358.4 nm) or ultraviolet sensitive (UVS, cone peak sensitivity from 355–380 nm and 50% of the incident light on the cornea transmitted to the retina as low as -323 nm)[6,7]. The UVS cone

[1]Ecology and Evolutionary Biology, School of Biosciences, University of Sheffield, Alfred Denny Building, Western Bank, Sheffield S10 2TN, UK. [2]Department of Computer Science, University of Sheffield, Regent Court, 211 Portobello, Sheffield S1 4DP, UK. [3]Bird Group, Department of Life Sciences, The Natural History Museum at Tring, Akeman Street, Tring HP23 6AP, UK. ✉e-mail: yhe20@sheffield.ac.uk; gavin.thomas@sheffield.ac.uk; c.cooney@sheffield.ac.uk

affords greater sensitivity to UV wavelengths as well as an enhanced ability to discriminate between colours. While absorption of UV is associated with darker skin pigmentation to aid photoprotection[8], UV reflectance is thought to be an important signalling mechanism in many bird species[9,10]. Although recent studies have advanced our understanding of the distribution of UV reflectance among bird species and on potential correlates[11–14], we lack a taxonomically broad and deep understanding of phylogenetic variation in UV reflectance and on how the combined effects of interspecific variation in visual system and light environment relates to the prevalence of UV in bird plumage.

We focus on the idea that UV may be important as a signalling channel. This leads to a series of predictions on (i) the ecology of UV reflectance and (ii) sex and body region differences in UV reflectance. Specifically, we predict that the prevalence of UV reflectance in bird plumage is higher in bird species that possess UVS visual systems, occur in regions with relatively high levels of solar UV irradiance, and occupy primarily wooded or forested habitats. These predictions are motivated by the expectation that ambient light conditions with proportionally high levels of UV should favour the use of UV signals for achieving conspicuousness[3]. While open habitats have the highest total UV irradiance, the relative UV irradiance, compared to other wavelengths, is often highest in woodland and forest canopy habitats leading to the prediction that UV reflectance is likely to be an efficient form of signalling in these habitats[3,5]. Solar radiation has previously been implicated in avian skin colouration in relation to photoprotection, suggesting that there is geographic variation in the strength of selection exerted by UV[8]. Our suggestion that relative UV irradiance may predict plumage reflectance has not, to our knowledge, been tested in the context of signalling. Numerous studies have discussed the likelihood of higher UV reflectance in species with UVS visual sensitivities[13,15–17] with as yet inconclusive evidence across broad sets of taxa. If UV reflectance is an important signalling route[18] then we would expect, on average, males to exhibit a greater degree of UV reflectance than females. We further predict that UV reflectance is more prevalent in ventral, rather than dorsal, body regions. This is because ventral (i.e. front-facing) body regions are generally, though not ubiquitously, thought to play a stronger role in sexual signalling than dorsal regions[19,20]. While some specific patches, such as the rump, may buck this trend, overall we expect UV to be higher in ventral regions.

Testing these predictions requires data on UV reflectance spanning species with variability in both visual system and light environment. Significant advances in our understanding of bird colouration have come from broad-scale studies that are limited to the human visual spectrum (i.e. excluding UV) (e.g. ref. 21), or include UV but are either phylogenetically limited or have sparse species sampling (e.g. refs. 22–25). However, capturing the variation to test our hypotheses requires applications of methods that capture UV reflectance across a phylogenetically broad and dense species sampling. Measuring or digitising specimens from natural history collections has become a critically important step in generating large-scale datasets in ecology and evolution[26–28]. However, processing of digitised data (e.g. specimen photographs) remains a significant and labour-intensive challenge. Deep learning, a subfield of machine learning and the state-of-the-art of many computer vision tasks, offers significant potential in ecology and evolution to unlock vast amounts of data[29,30]. Here, we describe the analysis of a dataset of calibrated images recording both visible and UV reflection that allows accurate measurements of colour. To address the processing challenge we test the efficacy of, and subsequently apply, deep learning algorithms to segment specimens and extract objective measurements of UV reflectance.

Segmentation allows measurements of the entire plumage (i.e. colour and pattern) for each specimen, facilitating measurement of multiple metrics relevant to our goal of testing the drivers of UV reflectance including mean, peak, and presence of UV colouration across the entire specimen. Segmentation is commonly used on biomedical images to separate focal regions such as cells, organs, and bones[31–33] and is also beginning to be used more widely on digitised natural history datasets[34,35]. However, to be a truly scalable solution for thousands to potentially millions of images, segmentation methods must provide reliable output. We assess the performance of several traditional computer vision-based segmentation methods (thresholding[36], region growing[37], Chan-Vese[38], and graph cut[39], more information of these four methods can be found in Supplementary Table 1) and compare them to semantic segmentation using deep neural networks, specifically the DeepLabv3+ architecture[40] which is from a semantic segmentation method family called DeepLab[41,42]. DeepLabv3+ is considered one of the best deep learning methods for segmentation, achieving 89% accuracy on PASCAL Visual Object Classes 2012 dataset (PASCAL VOC 2012) which includes thousands of segmented photos in 21 classes[43]. The main advantage of deep learning on image tasks (e.g. image segmentation, image classification) is the use of the convolutional neural network (CNN). CNN is the core deep neural network architecture for feature extraction from images[44,45], which takes images as input and extracts features using convolutional and pooling layers. Trained CNNs can make predictions for tasks such as image classification[44,46], pose estimation[47,48] and semantic segmentation[41,49] using extracted features.

Here, using deep learning image segmentation and phylogenetic comparative analyses we show that UV reflectance is widespread across passerine birds and is predicted by variables related to species' light environment and, to a lesser extent, visual system. To do this, we assess the performance of deep learning segmentation in comparison to classic computer vision methods using photos of bird specimens taken at the Natural History Museum, Tring, UK. We then test different methods to build a pipeline that can segment specimen photos automatically and accurately. We used, evaluated, and compared classic and deep learning segmentation methods to segment specimens from the background and to remove obstructions (labels, string etc.) that obscure the specimen in 5094 expert-segmented images. We then generated estimates of UV signalling in bird plumage using 146,652 images from 4545 passerine bird species using deep learning to (i) map the phylogenetic distribution of UV signalling and (ii) test how UV signalling relates to the visual system and light environment.

## Results
### Accuracy of deep learning for specimen segmentation
Across all three views, the DeepLabv3+ model achieved high intersection over union (IOU), precision, and recall scores (Fig. 1 and Supplementary Table 2). IOU captures the overall accuracy (combining elements of precision and recall), precision measures the proportion of correct predictions, and recall measures the proportion of true plumage area predicted by the model. Possible scores range from 0% to 100% for all three metrics (see 'Segmentation evaluation' in the Methods section for details). The mean IOU was 93.1% (per view, back: 94.6%; belly: 91.9%; side: 92.9%), and 88.8% of the segmentations (4525 out of 5094) had IOU higher than 90%. The lowest IOU is 53.6%. The mean precision was 96.3% (per view, back: 96.8%; belly: 95.7%; side: 96.4%) and 97.9% of the segmentations (4985 out of 5094) had precision higher than 90%. The lowest precision was 70.0%. The mean recall was 96.6% (per view, back: 97.6%; belly: 95.8%; side 96.2%). No segmentation had recall lower than 50%. Less than 0.2% of the results (7 out of 5094, per view, back: 1; belly: 4; side: 2) had recall lower than 75%, and less than 1.8% of the results (89 out of 5094, per view, back: 13; belly: 43; side: 33) had recall lower than 90%. Four out of the worst five segmentations were caused by low recalls and all have precision higher than 85%.

Figure 2 shows the predicted deep learning segmentations on a sample of images. Many examples correctly classified eyes and labels as non-plumage area (e.g. Fig. 2a, ii–iv). Three out of four (Fig. 2a, vi–viii) of the worst IOU segmentations were caused by low recall

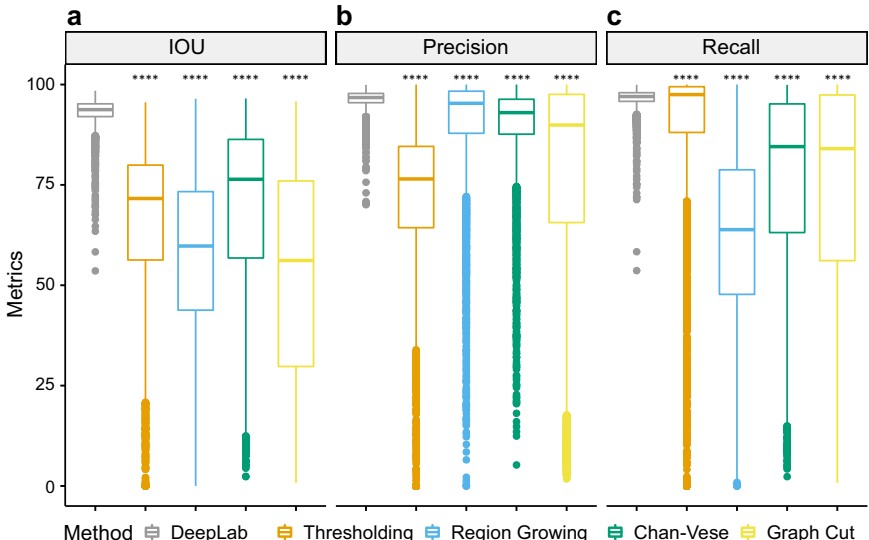

**Fig. 1 | The performance of predictions (N = 5094) from DeepLabv3+ and classic methods.** The tested classic methods are thresholding, region growing, Chan-Vese and graph cut. **a** IOU, **b** Precision, and **c** Recall are used to evaluate the performance. Asterisks indicate evidence for comparing the predictions of classic methods to the predictions of DeepLabv3 (ns: $p > 0.05$; *: $p \le 0.05$; **: $p \le 0.01$; ***: $p \le 0.001$; ****: $p \le 0.0001$). In box plots, a box indicates the median and first and third quartile, whiskers indicate range of data and points indicate outliers. Source data are provided as a Source data file.

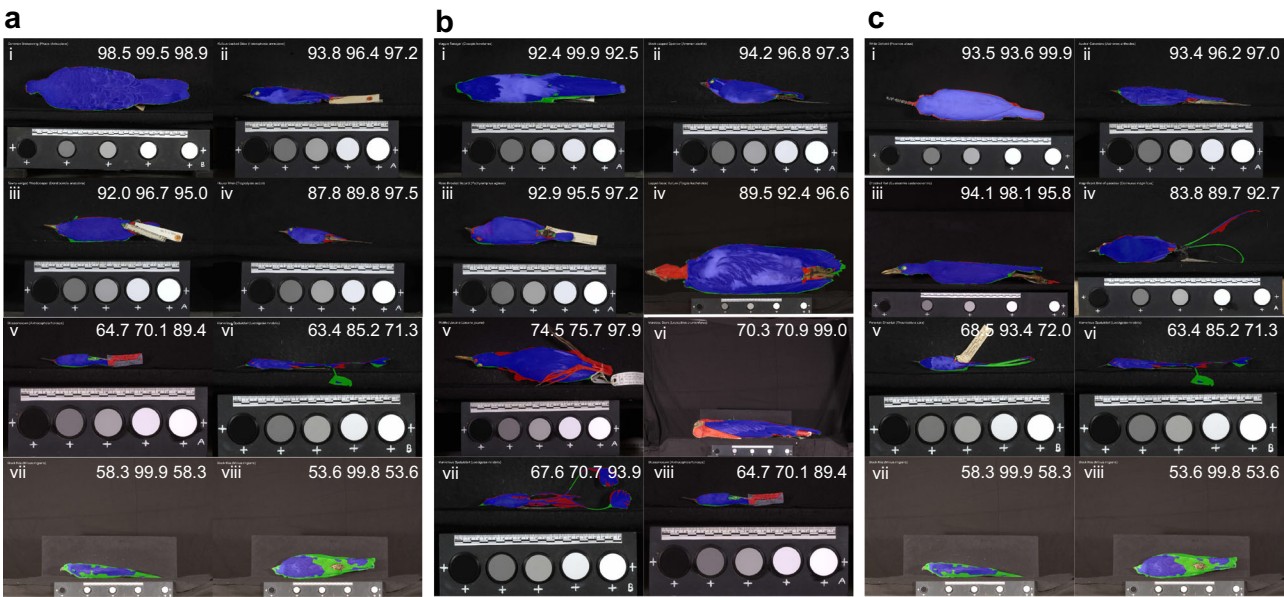

**Fig. 2 | Examples of images segmented by DeepLabv3+.** Images of the best, 50th, 75th and 95th percentile (ranked by metrics from high to low; from i to iv) and 4 worst predictions (from v to viii) based on **a** IOU, **b** precision and **c** recall. The IOU, precision and recall (from left to right) are displayed on the top right corner of each image. Blue is correctly predicted by the model (True positive); red is the non-plumage area that has been classified as plumage area by the model (False positive); green is the plumage area that has been classified as non-plumage area (False negative).

issues (shown as large green areas in Fig. 2). The two worst recall examples (Fig. 2c, vii–viii) had many undetected plumage areas, and these images have light black backgrounds and long camera distances due to the large size of the specimens. Other low recall examples failed to detect complete tails where tails are thin or irregular (Fig. 2c, v–vi). Thin tails can also cause low precision as the model misclassifies background surrounding thin tails as plumage area (Fig. 2b, vii). Legs that are placed on the top of the plumage area can be hard for the model to exclude (Fig. 2b, v). Figure 2b, vi, also shows an example of misclassifying an irregular beak as the plumage area.

Additional model testing: We found that (i) there was a significant effect of input resolution on accuracy where low resolution can result in low accuracy (Supplementary Fig. 1), (ii) the input channel using RGB has the highest performance (Supplementary Fig. 2), (iii) predictions using an augmented training set including randomly manipulated versions of existing images were marginally worse than predictions using the original training set (Supplementary Fig. 3), (iv) training models individually by view did not increase the accuracy (Supplementary Fig. 4), (v) low-quality datasets caused slightly lower model performance but the degradations were small (1–2%; Supplementary Fig. 5), (vi) the size of the training set is positively correlated with the model performance but DeepLabv3+ can achieve over 90% IOU, precision and recall using just 15% of the original training set (Supplementary Fig. 6), and (vii) that model performance was generally high

regardless of the level of contrast between the specimen and the background, but declined marginally with increasing contrast values, which appears to be driven in part by lower sample sizes at higher contrasts (Supplementary Fig. 7). The full details of these results can be found in Supplementary Note 1.

## Deep learning versus classic computer vision methods for specimen segmentation

We compared the results from the DeepLabv3+ model to four classic computer vision segmentation methods (thresholding, region growing, Chan-Vese and graph cut). IOU varied significantly among segmentation methods (ANOVA: F = 3141.3; d.f. = 4, 25465; $P < 0.01$), as did precision (ANOVA: F = 1678.6; d.f. = 4, 25465; $P < 0.01$) and recall (ANOVA: F = 1989.6; d.f. = 4, 25465; $P < 0.01$). DeepLabv3+ had superior performance for IOU, precision and recall compared to classic methods, combining both the highest mean values and lowest variance for each performance metric (Fig. 1), particularly when specimens have low contrast to the background (Supplementary Fig. 7). Specifically, DeepLabv3+ outperformed classic results by at least 23.4% on IOU, 6.4% on precision and 9.5% on recall. Graph cut had the best IOU among tested classic methods, while Chan-Vese had the best precision and Thresholding had the best recall. Graph cut was the overall best classic method in plumage images, while Chan-Vese segmented area conservatively, and thresholding tended to segment lots of non-plumage regions.

The worst examples from classic methods were clearly far worse than those from DeepLabv3+. Examples shown in Supplementary Fig. 8 illustrate that dark plumage, high plumage colour variability and museum specimen labels can be obstacles for classic methods whereas DeepLabv3+ segmented accurately on the same images.

## Phylogenetic distribution of UV colouration

Using manually inspected image masks produced by the DeepLabv3+ method, we mapped the phylogenetic distribution of UV colouration in passerine birds (Fig. 3). To do this, we converted plumage RGB pixel values into avian tetrahedral colour space[50] and derived three metrics capturing average (mean) and peak (upper quartile mean) relative UV reflectance (i.e. $u$ cone values), as well as a third metric (UV + colouration) designed to infer the presence of colours containing peaks of UV reflectance in combination with other colours (e.g. UV-yellow, UV-red). We found that, generally, UV reflectance represents a minor proportion of avian plumage colouration, with most plumages eliciting relative ultraviolet cone-catch values ($u$) of <0.25, where cone-catch values of 0.25 for all cones would be considered the achromatic null[50]. Despite this general pattern, the plumages of some species are characterised by high levels of 'pure' UV colouration, including the Purple Honeycreeper (*Cyanerpes caeruleus*) and the Hooded Mountain Tanager (*Buthraupis montana*) with peak dorsal $u$ values of 0.67 and 0.62, respectively, compared to a maximal value of 0.75. Using our alternative metric (UV + colouration) that accounts for the fact that UV reflectance may co-occur with reflectance at other wavelengths, we found more extensive evidence for UV colouration across passerines (Fig. 3), albeit with a similar pattern of phylogenetic clustering. Indeed, phylogenetic heritability ($H^2$) estimates[51] for the three UV reflectance metrics we consider were all >0.80 (range 0.81 to 0.93) (Supplementary Table 3), indicating that UV colouration—or a lack thereof—is phylogenetically conserved across passerines, with closely related species typically exhibiting similar levels of UV colouration.

## Correlates of UV colouration

We find that the degree of UV colouration is significantly predicted by several factors (Fig. 4, Supplementary Table 4). Specifically, for average and peak $u$, we find that values are significantly higher in males, on the dorsal side of the bird, in species inhabiting forests and particularly the upper strata and canopies of forests, and in locations with

relatively high incident UV radiation. Importantly, the positive effect of incident UV was independent of separate effects of total solar radiation and temperature effects on average and peak $u$. Our models also revealed a notable positive association between an ultraviolet sensitive (UVS) visual system and the degree of UV reflectance, but this effect was statistically non-significant and characterised by a high degree of parameter uncertainty (Fig. 4, Supplementary Table 4). Results based on the UV + colouration metric were similar to those based on mean and peak $u$ values, with the exception that (i) the effect of incident UV was no longer significant and (ii) UV + colouration is significantly more likely to be present on the ventral, not dorsal, side of the bird. Marginal $R^2$ estimates associated with these models ranged from 0.03 to 0.07 and results were generally consistent irrespective of the precise thresholds used to calculate metric values (Supplementary Table 4).

## Discussion

Our results show that UV reflectance, particularly as a component of other colours, is widespread across the passerine radiation, expanding and confirming inferences from previous studies[11,13,14]. Some clades [e.g. tanagers (family: Thraupidae), corvids (family: Corvidae), thrushes (family: Turdidae)] are particularly notable for the extent of UV reflectance whereas others have comparatively low incidence or prevalence [e.g. larks (family: Alaudidae), ovenbirds and woodcreepers (family: Furnariidae)]. Accordingly, the presence of UV in both male and female plumage shows a strong phylogenetic signal, in line with a relatively high degree of phylogenetic conservatism in the evolution of UV colouration in passerine birds[14].

Our results also provide further insight into the ecology of avian UV signalling. In particular, we find evidence to support Endler's[5] sensory drive hypothesis emphasising the role of light environment and habitat characteristics in shaping colour signal evolution. Specifically, our finding that UV is a more dominant component of plumage reflectance in species that (i) occur in regions with relatively high levels of incident UV solar radiation (i.e. controlling for correlated variation in temperature and total solar radiation) and (ii) are highly forest-dependent and/or canopy specialists. These patterns are in line with expectations that UV colours function well as conspicuous signals in locations proportionally rich in UV wavelengths and in habitats where UV contrasts well against background vegetation, given the prevailing light conditions[3,5]. Similar relationships between UV signalling and habitat characteristics (e.g. light environment) have been demonstrated previously[12,52] but our results suggest that such associations hold more generally across passerine birds. Previous studies have also highlighted the importance of humidity and temperature in driving macroecological variation in plumage colour with two alternative predictions. According to Gloger's rule, darker colours are associated with higher temperature and humidity and provide protection from solar radiation[53]. For example, Passarotto et al.[54] found higher degrees of melanism (darker and redder colours) in owl plumage towards the equator, consistent with Gloger's rule. Other studies have found support for a role of pigmentation in protection from UV radiation (e.g. ref. 8). Alternatively, Bogert's rule[55] predicts darker colours in colder climates because of the greater absorption of thermal energy. For example, Galván et al.[56] found support for the thermal niche hypotheses in a study of Iberian birds. Delhey et al.[57] show how these macroecological effects can interact and lead to more complex, but predictable, variation in plumage colour. Our results offer a perspective on the macroecology of colouration that links to the signalling environment. We suggest that solar radiation, and specifically relative UV, could also mediate global scale variation in colour.

We also find that UV reflectance is generally stronger in males than females and that average and peak UV reflectance is stronger on the dorsal rather than ventral side of the body. The stronger UV signal in males is in line with the idea that UV reflectance may play an important role in sexual signalling, involved in female choice and/or male-male

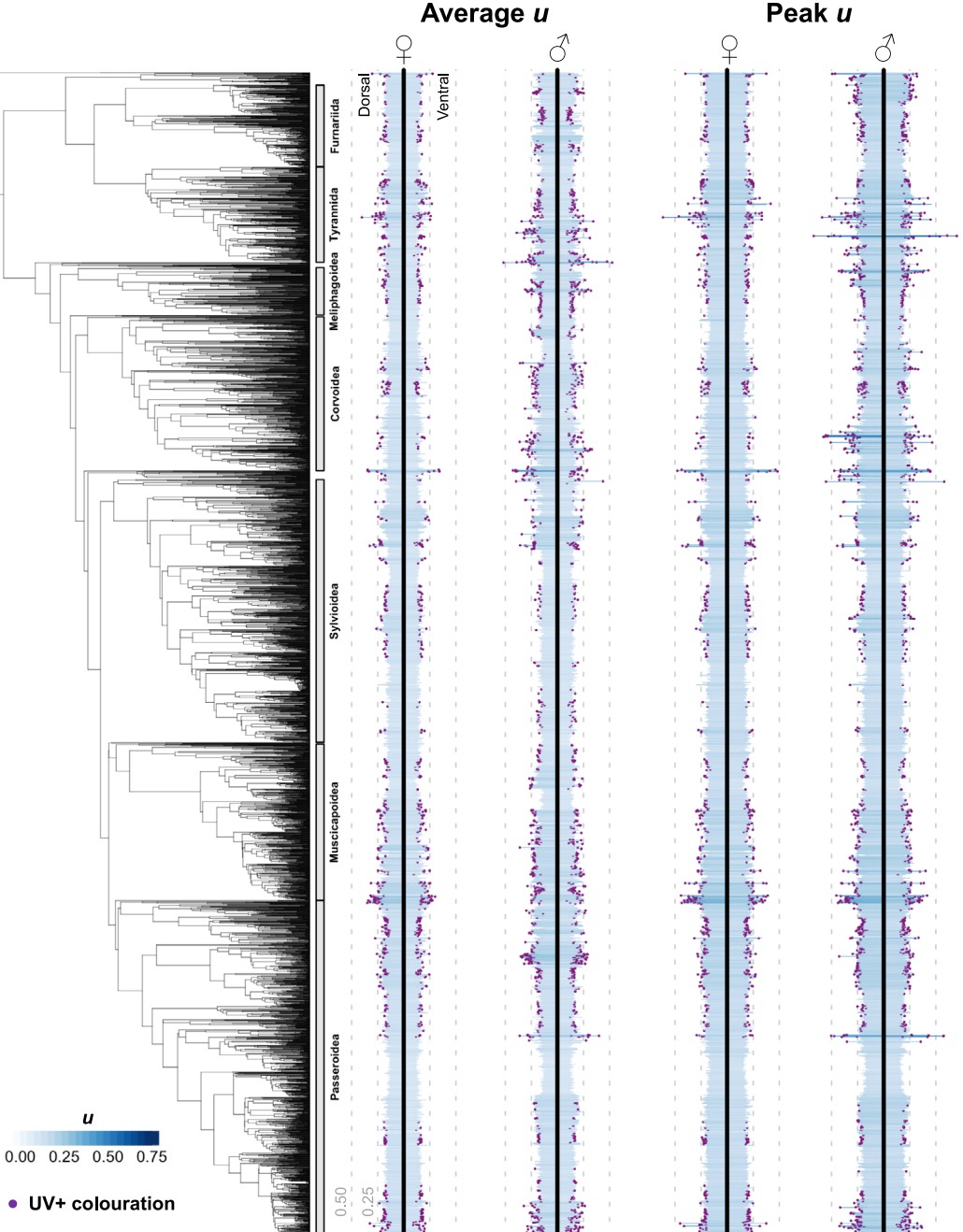

**Fig. 3 | The phylogenetic distribution of UV colouration in passerine birds.** Blue bars indicate the relative contribution of ultraviolet reflectance to plumage colouration (as measured by u values) of female and male individuals for 4545 species of passerine birds. Purple dots on the end of bars ('UV+ colouration') indicate the occurrence of detectable peaks in UV reflectance possibly occurring in combination with other colours (e.g. UV-yellow). Source data are provided as a Source data file.

competiton[18]. In contrast, finding that relative UV reflectance (i.e. average and peak *u* values) is generally higher on the dorsal rather than ventral side of the body is opposite to our prediction and is seemingly at odds with the idea that UV reflectance for signalling purposes should be concentrated on the front-facing (i.e. ventral) side of the body. Different bird body regions are likely to have different roles along a crypsis to conspicuousness spectrum and it is often suggested that dorsal body regions are under greater selection for crypsis than ventral regions[20]. Thus, we suggest that greater 'pure' UV reflection on dorsal regions may reflect the balance of selection for enhancing signal conspicuousness to conspecifics whilst minimising visual cues for potential predators, who may often be less visually sensitive to UV than passerines[9,10,58]. Further, it is interesting to note that we find the opposite pattern with respect to our UV + colouration metric, which indexes UV reflectance occurring in combination with other colours (e.g. UV-red, UV-yellow). Unlike levels of 'pure' UV reflectance which are higher dorsally, we find that UV + colouration is more common on ventral body regions, potentially indicating that UV may often act as a signal enhancer or amplifier in front-facing body regions that are often dominated by carotenoid-based colours[22]. For example, many conspicuously coloured passerine species, such as the Hooded Mountain Tanager (*Buthraupis montana*), tend to display colours rich in short wavelengths (e.g. structural blues, UV) on their dorsal regions and colours rich in longer wavelengths (e.g. carotenoid yellows, reds) on their ventral regions. While anecdotal, examples such as this suggest that this arrangement of colour is a potentially common solution to

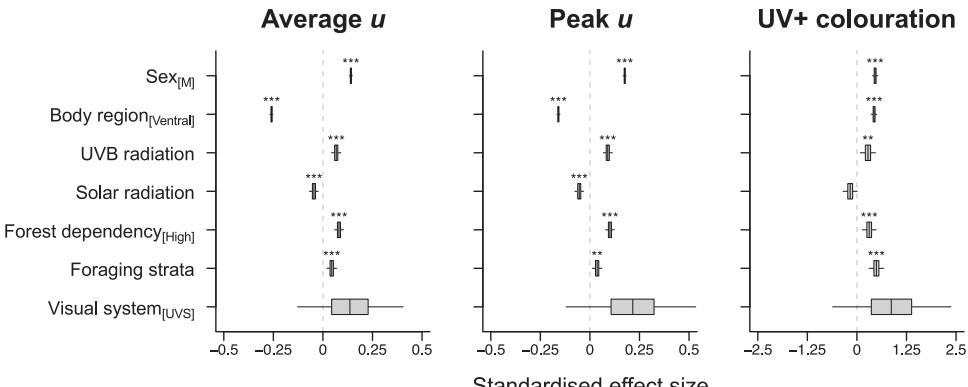

**Fig. 4 | Predictors of UV colouration in passerine birds.** Box plots summarise the posterior marginal distributions for all fixed-effects from Bayesian phylogenetic mixed models (two-sided tests, no adjustments for multiple tests) applied over a sample of 100 phylogenetic trees. Box widths represent the interquartile range, the median is shown as a vertical line within each box, and whiskers denote the 95% credibility interval of the distribution. Asterisks indicate evidence for a non-zero effect of the relevant variable. *P < 0.05; **P < 0.01; ***P < 0.001. M, male; UVS, ultraviolet sensitive. Peak u and UV+ colouration results correspond to thresholds of 25% and 5%, respectively. Results for other thresholds are given in Supplementary Table 4. Source data are provided as a Source data file.

maximising conspicuousness under various interacting selection pressures, though this suggestion remains to be explicitly tested.

Our analyses suggest a weak trend towards higher UV plumage reflectance among species with ultraviolet rather than violet-sensitive visual systems, in line with results based on visual modelling predicting only weak relationships between visual system variation and plumage colouration[17,59]. However, we are necessarily cautious in this interpretation because there is wide uncertainty in parameter estimates for the effect of VS/UVS in our models. This uncertainty likely stems from the relatively low number of transitions between visual systems, and conservatism within clades, in the passerine radiation[6]. However, it is also worth noting that visual system information is relatively sparse among passerine species and so we expect the empirical relationship between UV reflectance and visual system to become clearer as data for more species become available.

Taken together, our analyses reveal the diversity and extent of UV reflectance in passerine birds and provide insight into the factors that underpin the ubiquity of UV colours in the avian colour gamut. The data on which these inferences lie, rely on efficient processing of a vast quantity of raw input. We were able to achieve this using deep learning after first testing the suitability for these methods. We show how DeepLabv3+ can automatically segment bird plumage areas from other parts across more than 140,000 images within a few days on a consumer grade graphics processing unit (GPU, a processing unit that does calculations in parallel and is the key hardware for many Deep Learning algorithms) and can identify the plumage area (precision: 96.3%) and plumage area completeness (recall: 96.6%) reliably.

Deep Learning segmentation algorithms have developed very rapidly, and there are many algorithms that make accurate segmentations[41,49,60,61]. Here we used DeepLabv3+, which was shown to be the best algorithm for semantic segmentation tasks using the PASCAL 2012 benchmarking dataset until 2020. We note that a more recent algorithm, EfficientNet-L2 + NAS-FPN, that was published after our analyses were completed, appears to offer marginal improvements (1.5% improvement in mIOU) over DeepLabv3+ on the PASCAL 2012 dataset[61]. Future developments will likely further improve segmentation accuracy. We focused on one well-established algorithm in order to test the effectiveness of deep learning for segmenting the plumage dataset, providing a benchmark for future improvements and a comparison of the performance of deep learning and classic segmentation methods.

Our analysis showed that segmentation using DeepLabv3+ strongly outperformed all classic computer vision methods. Indeed, segmentations from classic methods are frequently so poor that they would often be unusable for downstream analyses of colour. Of the classic methods, graph cut had the best average plumage area IOU but was 23.4% worse than the average IOU from DeepLabv3+. In contrast to the DeepLabv3+ predictions, images with dark birds and prominent label tags could not be reliably segmented using classic methods. Dark birds were normally under or over segmented, and label tags were included as plumage area (e.g. Supplementary Fig. 8). Besides deficiencies shown in these examples, setting starting parameters for classic methods, for example, choosing threshold values for thresholding and region growing by hand-crafted image features, is a troublesome task[62,63]. We suggest that deep learning is likely to be of wider value for high throughput processing of very large image datasets and supports growing recognition of the potential value of deep learning for many applications in biodiversity science[29,30].

Our experimental configurations also allow us to identify limitations and possible ways to further improve model performance for deep learning. We found that input image resolution had positive effects on performance, as expected and previously reported for DeepLabv2[41]. In contrast, image augmentation, using additional channels and subsetting models did not improve the performance of the deep learning model. The best performance overall was achieved with DeepLabv3+ and an input resolution of 618 × 410 pixels. This resolution was the maximum we could achieve with available resources but could be increased with a more powerful GPU and we would expect that performance can therefore be improved further. Our results are also consistent with previous studies showing that the training set size is positively correlated to the model performance[64,65]. However, small training set sizes did not decrease the performance drastically. It is possible to use just 15% of the original dataset (~600 images) to generate segmentations with 90% IOU on 1018 validation images. This is still much more accurate than results using any of the classic method segmentation methods. The highly consistent imaging layout in our data may reduce the size of training data needed to get an acceptable result from deep learning.

The consistency of imaging in our data may partly explain the quality of performance of the deep learning model. The IOU in our best configuration was 93% which is higher than DeepLabv3+'s performance (mIOU: 89.0%) on the standard PASCAL VOC 2012 dataset[40]. In contrast to the PASCAL dataset, (i) our dataset has only two classes (plumage and non-plumage) while the PASCAL dataset has 21 classes[43] and (ii) our images consist of few and fixed focal objects (one) under a consistent, high resolution imaging setup. In contrast, the PASCAL images are more varied (e.g. different objects, backgrounds). While there are specific challenges in removing unwanted parts of the images

(including eyes and specimen labels), these do not seem to significantly impact model performance. These two factors may explain why no improvements were observed with image augmentation, additional channels and subsetting models, as the model had already been well trained on and fitted to a highly standardised original dataset.

Modern pipelines for museum collection digitisation typically follow similarly consistent standards such as uniform specimen placements, background and light environment[35,66,67] suggesting that such data can be analysed with deep learning. However, high standard digitisation is time-consuming. We applied our trained model to simulated low-quality images and it did not provide excessively inaccurate predictions, and the worst performance was much better than classic methods' results (i.e. Fig. 1, Supplementary Fig. 5a). Nonetheless, additional workflow steps to improve the consistency of training images (e.g. aligning images to a particular orientation) may be beneficial. However, overall this result, along with promising results on low consistent datasets such as PASCAL VOC 2012[40], shows that the DeepLabv3+ model is likely to be robust on less consistent datasets.

Here, we have tested and applied deep learning approaches for semantic segmentation to reveal the prevalence and predictors of UV plumage colouration across bird species. However, deep learning has broader potential applications for image processing including species identification and key point placement (e.g. landmarking for geometric morphometrics). Some tasks may require larger training sets than we have used. For example, DeepLabv2[41] used a training set size of 1400 images in PASCAL VOC 2012 and 2975 images in Cityscapes[68]. Tasks like classification and pose estimation have used even larger datasets, such as 1.2 million training images in ImageNet classification[69] and more than 28,000 images in MPII pose estimation challenge[70]. Such large training sets can be generated through citizen science projects, such as the 'Zen of Dragons' (https://www.zooniverse.org/projects/willkuhn/zen-of-dragons). Regardless of the source of training data, all automated methods are likely to be imperfect and, depending on the goal of the project, may require expert error checking prior to downstream analysis as we used here. Nonetheless, we support the view that deep learning has great promise[29,30]—particularly in the mobilisation of digitised images (both 2D and 3D) from natural history collections—allowing new data to be brought to bear on key outstanding questions in ecology and evolution.

## Methods

### Specimen imaging

The specimen image and label data used in this study were taken in the bird collections at the Natural History Museum, Tring, and is approved by the institution for use in this work. All images followed a standardised design[22]. Photos were taken from three views (back, belly and side) for each specimen and each view was photographed twice, once in the human-visible and once in the ultraviolet (UV) light spectra, enabled by using a Nikon 105 mm f/4.5 UV Nikkor lens and a modified Nikon D7000 DSLR camera. The camera was modified (by Advanced Camera Services, Norfolk; http://advancedcameraservices.co.uk/) to allow both human visible and ultraviolet (UV) wavelengths of light to be recorded. For each view, pairs of images (human-visible and UV) were taken in the human-visible or UV spectrum by using either a Baader UV/IR Cut filter/L filter (transmits light in the human visible range 400–680 nm) or a Baader U-Venus-Filter (transmits light in the UV range 320–380 nm). Each image included one specimen and a set of five Labsphere Spectralon diffuse reflectance standards (2%, 40%, 60%, 80% and 99% reflectance, arranged left to right in each image, referred to as Standard 1–5) photographed against a non-reflective black background (theatre blackout curtains) under controlled lighting conditions (two Bronocolor Pulso G 1600 J lamps with UV filters removed and powered by a Broncolor Scoro 1600S Power Pack).

Specimens were placed with heads on the left and tails on the right in images where possible. Due to variation in size and shape of different species (e.g. exceptionally long neck or legs) some museum specimens are arranged in non-standard ways (e.g. fold necks to fit specimens in the camera). The same camera settings were used for all photographs (1/250 s, f/16.0, 'Daylight' white balance, RAW photo format), with the exception that ISO was 100 for human visible images and 1000 for UV images. Images were saved in RAW format at a resolution of 4948 × 3280 pixels. The dataset used for training and validating the deep learning models consists of 5094 images (visible light only) from 1698 specimens and species. The dataset predicted by deep learning and used in UV analysis consists of 146,652 images (both visible and UV light) from 24,442 specimens and 4545 species.

### Image segmentation with deep learning

We used DeepLabv3+[40] to create a segmentation workflow with two steps: (i) data preparation, including expert labelling to generate training and model evaluation datasets, and image downsampling; and (ii) model training and application.

**Data preparation.** We produced data for model training and assessment by manually labelling a subset of 5094 photos representing three views of 1698 bird species. The sample of 1698 bird species encompass representatives of more than 81% of bird genera and 27 bird orders, so the labelled images capture a large extent of the total variation in plumage colour, patterns, and bird body shape. Examples of expert labelling are shown in Fig. 5. We used multiple polygons to capture unconnected areas (Fig. 5b) and nested polygons to label non-plumage areas inside plumage areas (e.g. eyes and feet; Fig. 5c). Our goal is that segmentation should not include any regions outside the plumage area, and it is preferable to segment within the focal area (i.e. to be conservative in the estimation of the plumage area) to ensure that the colour space only contains plumage colour information. The resulting manual segmentation then contains two classes: plumage areas and non-plumage areas.

DeepLabv3+ outputs heatmap arrays in which the array resolution is the same as the input image and where the number of matrices in the array is equal to the number of pixel classes. Here, we have two-pixel classes distinguishing pixels that are either inside or outside the segmented area. The output heatmap pixel value (0 to 1) of each channel represents the probability that the pixel belongs to the corresponding class. We converted coordinates of expert drawn polygons to heatmaps, with the first channel as the non-plumage area and the second channel as the plumage area. Pixels of the non-plumage area were set to 1 for the first channel and 0 for the second channel, and vice versa for pixels of the plumage area.

The DeepLabv3+ architecture is most efficient when run on a GPU but typically requires downsampling of input images to avoid memory limitations. We used a 12GB NVIDIA GTX 1080Ti GPU and downsampled all 5094 images to 618 × 410 pixels (from 4948 × 3280 pixels) using bilinear interpolation from the OpenCV computer vision library[71]. This resolution is eight times smaller than the original resolution and is the largest resolution that could be trained within memory limitations.

A common approach used in many studies is to split data into a training set, a validation set and a test set that is used to provide the final benchmark (e.g. methods used in solving the ImageNet challenge[44,45]). Here, we used only training and validation sets so that every image from the labelled dataset (covering a wide range of extant bird species) has a prediction from the same data partition routine. This allows the relationship between bird taxonomy and network performance to be evaluated (i.e. to assess whether performance varies among bird clades due to broad differences in size, shape and colouration of specimens). We split the 5094 expert labelled images into a training set and a validation set with an 80:20 ratio.

**Fig. 5 | Examples of using polygons to segment plumage areas of specimens. a** A specimen is segmented using a single polygon. **b** A specimen is segmented using multiple polygons. **c** A specimen is segmented using nested polygons as the eye is not plumage area and is excluded using a nested polygon.

**Model training and application.** After data splitting, we trained the model with the training set (80% of images) under a set of pre-defined network hyperparameters (see below). We used five-fold cross-validation to provide an accurate estimate of model performance by averaging performance for different partitions (five partitions for five-fold cross-validation) of training and mutually exclusive validation sets. For each training step, the network generates predictions from input images. The model optimises the loss between output heatmaps and ground truth heatmaps (i.e. the expert labelled validation dataset) by updating its parameters with the gradient of a loss function[72]. We used the sum of cross-entropy between pixel values of output heatmaps and ground truth heatmaps as the loss function[41]. To minimise the loss function, we used the ADAM optimiser[73] and the gradient of the loss function to update model parameters. We set the initial learning rate to 0.01. Through the training process, the learning rate was cosine decayed and restarted at the initial value after reaching zero, which increases the likelihood of reaching a better local optimum[74]. The length of the first period of decay-restart was set to one epoch (defined as one pass of the full training set for the network). After each period, the new period is two times longer than the previous one (i.e. the second period takes two epochs to decay to zero, the third period takes four epochs and so on). We trained the model over 31 epochs (i.e. five complete decay-restart periods), after which the optimisation had converged (i.e. the loss has stopped decreasing).

We implemented and trained the network using Python 3 and the deep learning library Tensorflow (version 1.12)[75] on one NVIDIA GTX 1080Ti GPU (12GB GPU memory). The code can be found on GitHub (https://github.com/EchanHe/DL_seg_avian_plumage)[76]. To balance the memory usage of the GPU and the optimisation at each step[77] we divided training images into batches of four images. The model takes one batch per training step.

After the training process, we passed the validation images into the trained network to generate validation predictions. We then resized the predicted segmentations to the same resolution as the original images (4948 × 3280 pixels) and used these resized predictions, compared to the ground truth validation set to evaluate model performance (see below).

**Additional model testing.** In addition to the core pipeline above we also trained and validated the models (i) with alternative input resolution, (ii) with alternative input channels (human visible and UV), (iii) by applying image augmentation (a method that creates extra training images by manipulating the existing training images), (iv) by restricting training by image view (back, belly and side), (v) by lowering image quality, (vi) by adjusting the size of the training set to test if these effected the performance of the DeepLabv3+ model, and (vii) by exploring how the contrast between plumage and non-plumage areas within images affected model performance. The details of these tests and results can be found in Supplementary Note 1. In the main text, we focus on the core pipeline outlined above since this proved to be the best model configuration in our evaluation tests.

**Image segmentation with classic computer vision methods**
To test if deep learning outperforms classic computer vision methods on this dataset, we also tested the performance of the thresholding, region growing and graph cut methods from the OpenCV library[71] and Chan-Vese from the scikit-image library[78]. A weakness of some of these methods is that while they do not require any prior knowledge of the shape of the segmentation area, the region growing, Chan-Vese, and graph cut (but not thresholding) methods do require spatial information as starting values (see Supplementary Table 1). These are usually points within the focal region. We used points within the body region (2D points that are placed on specific bird body regions) as initial spatial information. We applied gaussian smoothing, a common pre-processing step to reduce noise for many classic segmentation methods[79], prior to applying each of the four classic segmentation methods outlined below. We applied morphological close (close segmentation holes) and open (remove segmentation noises) as a global post-processing step[80].

**Thresholding.** Thresholding segments an image by allocating each pixel to either the foreground or the background based on a pre-defined value[36]. This value can be set either manually or automatically calculated based on image features such as the image histogram or entropy[81,82]. For thresholding, we first converted images to greyscale. Along with segmenting the plumage area, thresholding will inevitably segment parts of the reflectance standards, as standards necessarily span the majority of greyscale values. We therefore reduced the target area by selecting the most upper connected component of the image. This is possible because the specimen is always placed above the reflectance standards but requires the assumption that the segmented plumage area is not connected with other segmented parts. We tested whether using the modal pixel value of the image with a positive offset of 15 performs better than Otsu's[81] method and adaptive thresholding methods. We therefore used the modal pixel value to threshold images.

**Region growing.** Region growing is a method for segmenting the neighbouring pixels of an initial pixel. The classification of each neighbouring pixel depends on its similarity to the initial pixel values. Region growing methods iterate the same procedure by examining the neighbour pixels of newly segmented pixels until no more pixels can be segmented[37]. We tried 150 ranges from different upper (even numbers from 2 to 30) and lower (even numbers from 2 to 20) boundaries for region growing. We found that the best combination is a lower boundary of 6 and an upper boundary of 30 and we use these settings for evaluation and comparison to DeepLabv3+.

**Chan-Vese algorithm.** The Chan-Vese algorithm is an active contour model designed to detect object outlines that are not defined by a gradient[38] and is a development of the 'snakes' active contour models[83]. The model requires a starting area within the segmentation area, which we initiated using squares of 20 × 20 pixels around points placed on the specimen and applied the algorithm for 100 iterations[38].

**Graph cut**. The graph cut algorithm[39] treats an image as a graph where pixels are nodes. Each pixel has edges to its neighbour pixels, and edges to a source (foreground) and a sink (background) node. Weights of edges are based on pixel intensities and identities (i.e. foreground, background or to be segmented). The minimum cut cuts the graph into two subgraphs that have the largest weighted sum[39]. The result is the foreground subgraph defining the segmented object. For the graph cut method, we set points placed on the specimen as the foreground. The consistent setup for imaging specimens means that specimens would not be placed near the top, bottom, left and right boundaries, and would always be placed above the reflectance standards. We therefore set pixels within 20 pixels of the top, left and right edges and below the standard points as background.

### Segmentation evaluation

We used a range of metrics to evaluate the performance of both the deep learning and classic computer vision models. These focused on capturing the precision (positive predictive value) and recall (sensitivity) of the segmented areas and on assessing the reliability of colour information extracted from the segmentations. To assess the segmented areas we used the mean intersection over union (mIOU), precision, and recall metrics. The mIOU is the average IOU of all classes (e.g. plumage area and non-plumage area for the dataset). The IOU of class $i$ is:

$$\mathrm{IOU}_i = \frac{p_{ii}}{p_{ii} + p_{ij} + p_{ji}},\qquad(1)$$

where $p_{ii}$ are pixels of class $i$ and classified as class $i$ (true positive); $p_{ij}$ are pixels of class $i$ but classified as other classes (false negative); and $p_{ji}$ are pixels of other classes classified as class $i$ (false positive). IOU is a straightforward metric to measure the segmentation performance by combining aspects of both precision and recall but it can be useful to consider precision and recall separately. Precision shows the proportion of correct predictions and is a useful test of predictive capability of the model, whereas recall measures the segmentation area that the model does not predict and reflects sensitivity of the model. We used the following formulas for precision and recall of class $i$:

$$\mathrm{Precision}_i = \frac{p_{ii}}{p_{ii} + p_{ji}},\qquad(2)$$

$$\mathrm{Recall}_i = \frac{p_{ii}}{p_{ii} + p_{ij}},\qquad(3)$$

We used IOU and precision to measure the network performance as they both reflect the project-specific goal of minimising the inclusion of non-plumage regions of the image. Achieving high recall is less critical but nonetheless important because we do not want results with excessively low recall (i.e. that are conservative). Segmentations have only two classes (plumage area and non-plumage area) that are mutually exclusive, so mean metrics and plumage area metrics are highly correlated. We therefore report metrics based on the evaluation of the plumage area only.

### UV data and analysis

**Image processing.** Focusing on passerine species with male and female data, all raw (.NEF) images of specimens were linearised and exported as linear TIFF files using DCRAW[84]. Following established approaches[22,85], pixel values were normalised using mean pixel intensity values from the five grey standards included in each image in order to control for variation in lighting conditions. We then segmented images using the image masks described above to leave only pixel values corresponding to the specimen in each image. Importantly, prior to pixel extraction each image mask was individually checked by

eye and manually refined where necessary using bespoke software (https://github.com/EchanHe/PhenoLearn)[86]. The final dataset consisted of images for 24,442 specimens covering 4545 passerine species, with an average of 2.8 male and 2.6 female specimens per species.

As individual pixel values can be noisy, and because different specimens were represented by different numbers of pixels due to their relative size in the image, we downsampled specimen images to a comparable resolution prior to extracting data on UV reflectance. To do this, we treated each specimen image as a raster and used the aggregate() function in the R package 'raster' (version 3.4-5)[87] to find the smallest aggregation factor in the range 100 to 1 that resulted in at least 500 aggregated cells (pixels) being returned. We then randomly sampled 500 observations from this aggregated dataset to represent the plumage colouration for a particular specimen view in all further analyses.

**Visual modelling.** We used methods developed by Troscianko and Stevens[85] to generate mapping functions to convert sampled specimen RGB pixel values into avian cone-catch values. Using tools available in the IMAGEJ Multispectral Image Calibration and Analysis Toolbox (version 2.2; http://www.empiricalimaging.com/), we generated mapping functions for each photoreceptor using equations containing second-order polynomial terms and three-way interactions between channels. Note that this approach does not incorporate information on camera responses in the UV from the camera's green channel due to typically low sensitivities of the G channel in the UV range[85]. We fit these equations to our data incorporating information on the estimated spectral sensitivities of our camera set-up and the irradiance spectrum of our illuminant (i.e. flash units), both of which we estimated previously[22]. For modelling receptor responses, we assumed idealised illumination conditions[25,50] and receptor sensitivities corresponding to an average ultraviolet-sensitive (UVS) avian visual system, extracted from the R package 'pavo' (version 2.6.1)[88]. We used this information to generate mapping functions for each cone class, and the resulting models were all characterised by a high degree of mapping accuracy ($R^2$ values >0.99). These mapping functions were used to estimate relative cone-catch values ($u$, $s$, $m$, $l$), which measure the relative contribution of ultraviolet ($u$), shortwave ($s$), mediumwave ($m$) and longwave ($l$) reflectance to plumage colour[25,50]. Our previous work has demonstrated that cone-catch values generated by this photography-based approach are highly correlated ($r > 0.92$) with corresponding values calculated from spectrophotometric measurements[22]. Finally, as quantifying the colour of patches with low overall reflectance can be problematic[52], pixels exhibiting a mean normalised reflectance value of <1% across all channels were re-cited to the achromatic centre (i.e. $u = s = m = l = 0.25$).

**UV colouration metrics.** We considered three metrics for quantifying differences in UV colouration: two based on variation in $u$ values across plumages[50] and a third based on determining the presence of colours containing peaks of UV reflectance that may also stimulate other cone types (e.g. UV-yellow, UV-red)[10,52].

First, we calculated the average (mean) and peak (upper 25% mean) $u$ values for each image. Values of $u$ provide a tetrachromatic estimate of the ultraviolet contribution to plumage colouration[50] and whereas mean $u$ values quantify the average UV reflectance across whole plumages, peak $u$ values are suitable for capturing the UV reflectance of smaller patches of colour. Second, we employed a different approach to inferring the presence of UV colouration that involves identifying colours containing a UV peak (or a peak encompassing the UV range) but that may also stimulate other colour cones[52]. Specifically, following ref. 52, we categorise colours as 'UV+ colouration' if reflectance measurements satisfy three criteria: (1) $u$ cone sensitivity shows a quantum catch higher than 0.05 relative to a theoretical maximum 100% white reflectance standard, (2) reflectance

over 300–400 nm exceeds 3% reflectance, and (3) reflectance over 300–400 nm is higher on average than the minimal reflectance over the range 400–700 nm. The latter criterion is crucial as this allows us to identify cases in which peaks of reflectance occur in the UV, even reflectance at other wavelengths dominates. We applied these criteria to all pixel values in an image and counted the number of pixels (out of 500) subsequently categorised as a UV colour. We considered a specimen image to have evidence of UV colouration if >5% of pixels were categorised as having UV colouration. We also assessed the sensitivity of our results by investigating alternative thresholds used to calculate peak $u$ and UV+ colouration metrics. Specifically, in addition to the thresholds outlined above, we (i) calculated peak $u$ values based on the upper 50% and 10% of $u$ values and (ii) considered UV+ colouration to be present using thresholds of >1% and >10% of pixels.

Overall, we note that the first two metrics measure the degree to which plumage colouration exclusively stimulates the $u$ cone (i.e. represents 'pure' UV colouration), whereas the third metric (UV colouration presence/absence) maps the occurrence of detectable peaks in UV reflectance that may occur in combination with reflectance at other wavelengths (e.g. caused by carotenoid pigmentation). We calculated estimates of each metric for each image separately, and then calculated sex-specific, species-level values for each view (i.e. body region) as the average of specimen-level values. As side view images contained large areas of plumage already captured by back and belly images (e.g. Fig. 5), we restricted our analyses to back and belly (i.e. dorsal and ventral) views only, to minimise the risk of including the same plumage area twice in our analyses.

**Phylogenetic framework.** To provide a phylogenetic framework for the passerine species included in our analysis ($n = 4545$), we downloaded 100 trees from the posterior distribution of complete Hackett-backbone trees produced by Jetz et al.[89] from http://www.birdtree.org. These trees were then pruned to generate a distribution of trees containing only the focal species set. All of our comparative analyses were run over this distribution of 100 trees to incorporate phylogenetic uncertainty into our parameter estimates. For plotting purposes, we identified a maximum clade credibility (MCC) tree from this posterior distribution of trees using the maxCladeCred() function in the R package 'phangorn' (version 2.5.5)[90].

**Predictor variables.** To test the role of factors hypothesised to influence the evolution of UV plumage colouration, we collected data for three key variables: ultraviolet-B (UVB) radiation, the degree of forest dependency, and species visual system. In total, we were able to collect data on these variables for 4519 of the 4545 species in our dataset.

We used global data on UVB radiation as a proxy for UVA. UVA and UVB have highly similar global distributions[91] and we use UVB due to the availability of data designed for macroecological studies. Global spatial information on annual mean UVB radiation was extracted from Beckmann et al.[92] at 15 arc-minute resolution. To generate species-level values, we intersected this dataset with information on species' geographic ranges provided by BirdLife International (http://www.datazone.birdlife.org). To do this we first resolved taxonomic differences between the BirdLife and Jetz et al. datasets as far as possible, manually editing (i.e. combining or splitting) range maps for BirdLife taxa where necessary. We focused on species' breeding geographic ranges only (seasonality = 1 or 2) and regions where species are known to be native or reintroduced (origin = 1 or 2) and extant or probably extant (presence = 1 or 2). We extracted species' polygon range maps onto an equal area grid (Behrmann projection) at 0.5° resolution (~50 km at the equator) and then reprojected and resampled the UVB dataset to match the resolution of our range data. Species-level UVB values represent averages across their geographic range. Spatial variation in UVB is correlated with variation in total solar radiation and mean temperature[92]. Therefore to isolate the effect of (relative) UVB

irradiance on colouration by controlling for the potential effects of these variables in our analysis, we used data from the WorldClim database[93] (v2.1) (https://worldclim.org/) to generate similar variables capturing variation in terms of annual mean temperature (bio1) and total annual solar radiation across species.

Forest dependency information was extracted from BirdLife International's Data Zone (http://www.datazone.birdlife.org) and recoded as a binary variable to facilitate effect size comparison. Specifically, species were coded as highly forest-dependent ('medium' or 'high' dependency) or not ('low' dependency or 'does not usually occur in forest'). In a small number of cases ($n = 52$) we filled gaps in this variable by consulting species' records on http://www.birdsoftheworld.org. Species' foraging strata information was extracted from EltonTraits[94] and for each species we summed the proportion of time spent foraging in mid-high and canopy strata to given an index of the relative time spent foraging in the upper strata of highly vegetated habitats.

Finally, we categorised species as having a violet-sensitive (VS) or ultraviolet-sensitive (UVS) visual system primarily using the information presented in Ödeen et al.[95]. This dataset provides approximately family-level resolution on visual system variation across passerines, and using this information we coded lineages and their constituent species as either VS or UVS based on the available data. Where species/clades were not sampled by Ödeen et al.[95], we assumed that the visual system was the same as that of closely related lineages and/or the common ancestor. This makes sense as evolutionary switches between VS and UVS visual systems appear to be relatively rare across passerines[95] and birds more generally[6]. However, one exception to this rule appears to be in the Maluridae (Australasian fairywrens and allies), where multiple shifts between violet and ultraviolet vision have occurred within a single genus (*Malurus*)[15]. Therefore, for this genus, we used additional information[15] to recode species as necessary.

**Statistical analyses.** To test the relationship between UV colouration metrics and predictor variables across species, we used Bayesian phylogenetic mixed models implemented in the R package 'MCMCglmm' (version 2.32)[96,97]. All models were run over a posterior distribution of 100 trees to incorporate phylogenetic uncertainty and posterior distributions of parameter estimates associated with different trees were pooled to give model estimates that incorporate phylogenetic error[98]. In all cases, models were run for 110,000 iterations (sampled every 25th iteration) with a 10,000-iteration burn-in. Mean $u$ and peak $u$ were $\log_{10}$-transformed prior to model fitting and all variables except UV colour presence/absence were standardised (mean = 0, standard deviation = 1) prior to model fitting to facilitate effect size comparison. For continuous response variables (mean $u$ and peak $u$) we used family = "gaussian", whereas for our binary response variable (UV colouration presence/absence) we used family = "categorical". Correspondingly, we used two sets of standard non-informative priors: list(R = list(V = 1, nu = 0.002), G = list(G1 = list(V = 1, nu = 0.002)))] for gaussian models and list(R=list(V=1, fix = 1), G = list(G1 = list(V = 1, nu = 0.002))) for categorical models. Marginal and conditional $R^2$ values for each model were calculated using established methods[99]. Finally, phylogenetic heritability ($H^2$) values[51] were estimated by fitting intercept-only models for each variable of interest and then calculating the proportion of the total variance explained by phylogenetic effects across the posterior distribution of parameter estimates.

**Reporting summary**
Further information on research design is available in the Nature Research Reporting Summary linked to this article.

## Data availability
Specimen images, expert labelled images and machine-predicted coordinates associated with training and testing our deep learning model are available at figshare (https://doi.org/10.15131/shef.data.

19221699). All analysis data are available in the manuscript or in Supplementary information. A demo dataset and the analysis code are available on GitHub (https://github.com/EchanHe/DL_seg_avian_plumage) and archived at Zenodo (https://doi.org/10.5281/zenodo.6916988)[76]. The data used in the figures of this study are provided in Source data. Source data are provided with this paper.

## Code availability

Scripts for analyses are available on GitHub (https://github.com/EchanHe/DL_seg_avian_plumage) and archived at Zenodo (https://doi.org/10.5281/zenodo.6916988)[76].

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

## Acknowledgements

We thank M. Adams, H. van Grouw, R. Prys-Jones and A. Bond from the Bird Group at the NHM, Tring for providing access to and expertise in the collection. This work was funded by a Leverhulme Early Career Fellowship (ECF-2018-101) and Natural Environment Research Council Independent Research Fellowship (NE/T01105X/1) to C.R.C, and a European Research Council grant (615709, Project 'ToLERates') and Royal Society University Research Fellowship (UF120016, URF\R\180006) to G.H.T.

## Author contributions

Y.H., S.M., C.R.C. and G.H.T. designed the research; C.R.C., Z.K.V., L.O.N., C.J.A.M. and M.D.J. collected the image data; Y.H. and C.R.C. conducted the analyses. Y.H. wrote the manuscript, with input from all authors.

## Competing interests

The authors declare no competing interests.
