## [Peer Review File · Nature Communications]

Deep learning image segmentation reveals patterns of UV reflectance evolution in passerine birdsReviewers' Comments:

Reviewer #1:

Remarks to the Author:

This is a well-written and clear manuscript that describes an application of segmentation to understand the evolution of UV plumage. There are some major areas that need to be addressed, however.

-It was unclear to me whether the dataset images are available (both originals and "expert-labeled") - they must be made public so that other groups can use them to either reproduce the work or use them with other ways of performing segmentation.

-I think it's unfortunate that the method doesn't seem particularly robust to images taken differently and to rotated specimens. It seems like rotation of specimens could be corrected for in an image processing step in the workflow to mitigate this problem. The authors do spend some time exploring image augmentation, but it is still unclear to me why the results would be worse when image augmentation is implemented. I think the authors should spend some more time explaining why this was the case. Did they try all the potential augmentations offered by DeepLab?

-The authors use DeepLabv3+, but do not discuss other potential segmentation frameworks in detail. I think it is important that they describe why they chose DeepLab over other methods, what advantages and disadvantages it has over other methods. Because many researchers are just starting to get into deep learning without deep knowledge of the frameworks and methods, more background on the options and pros and cons of different approaches is necessary.

-I think there should be more discussion of prior research on UV coloration in birds, even if in much smaller scale than the work presented here. There is not much discussion about how this maps onto the phylogeny, whether there are surprises in particular clades, whether there are other common traits that appear to be correlated. While the segmentation work itself is interesting, the connection to the evolutionary story is essential and somewhat lacking here.

Small comments:

-In the text, DeepLab is referred to in a few ways, and because there are multiple versions of the software, this should be clarified, whether it is DeepLab, DeepLabv3, etc.

-In line 71, "and that allows objective measurements of colour." Deep learning models are not objective - this should be made clear.

Reviewer #2:

Remarks to the Author:

This is an exciting paper that represents a significant advance in the field. I really like the deep learning approach to extracting data from numerous images and the comparison with other approaches is insightful and provides strong justification for the use of the deep learning data over classical segmentation methods for the biological analysis. The authors conclude that UV-signalling in avian plumage is widespread, phylogenetically conserved, sexually dimorphic, dorsally predominant and associated with the light environment (UVB and forest dependency). It is well written, and the analysis is sophisticated and state-of-the-art.

I do not have any major concerns with this paper; however I make a few comments below for the authors to consider in a revision.

1. Lines 48 to 49. Worth mentioning here the specific wavelengths that the UVS cone is sensitive to. Although it is widely known in the field that human visible light is ~ 400 to 700 nm, and for some birds it can range from ~ 320 to 700 , it might be worth describing this around here in more detail for a more general audience.

2. Lines 54-58. Predictions. Include predictions about body regions and sexual dimorphism?

3. Lines 97 – changes “100’s of thousands” to the actual number.

4. Lines 178-180 and elsewhere in the paper. The authors mention a few times how UV reflectance “may” co-occur with reflectance at other wavelengths and I was curious about how birds with strong UV signals may appear to us? Is UV reflectance associated with violet and blue birds for instance. A figure with Average u, Peak u, and UV colouration vs reflectance from the wavelengths associated with the S, M, and L cones should be possible with their data set. This could provide insight into the private channel hypothesis mentioned in the discussion (see below). Additional mention of this is in Fig 3, where the authors state “Purple dots on the end of bars (‘UV colouration’) indicate the occurrence of detectable peaks in UV reflectance possibly occurring in combination with other colours (e.g. UV-yellow).”. I might be missing something here, but I do not understand why this can not be tested more specifically.

5. Lines 218-222 and first part of next paragraph. Following up from the last comment, it is noteworthy that the examples of clades with notable extent of UV reflectance (tanagers, corvids, and thrushes) are highly colourful overall in the Vis-channels, and the ones with low UV reflectance (larks, and woodcreeper/ovenbirds) are generally pretty drab and brownish. In the next paragraph the authors appear to argue that the higher extent of UV on the dorsal regions are consistent with the private channel hypothesis (lines 235-238. But I think an important part of the story here has not been explored in enough detail – in that UV signalling may be associated with high conspicuousness in the Vis-light channels. If that is the case, then it would be unsupportive of the private channel hypothesis because the UV-signals will also come along with highly visible colours that would still be detectable to eavesdropping receivers like predators. The authors can test this directly: Is UV a special secret channel, or is it simply an additional way for birds to be colourful, (that happens to be invisible to humans but that we can sort of still infer in the RGB channels)?

Line 252 – I got tripped up by this sentence – maybe insert comma between “lie” and “rely”

Lines 338 and elsewhere. The plumage UV reflectance is measured across 320-380nm by the Baader U-Venus-Filtre, and these wavelengths will be visible to birds with the UVS cone and correspond to UVA light (~ 320 - 400 nm). But one of the key environmental predictors used is annual mean UVB radiation (~ 280 - 320 nm) – and it is found to be associated with UVA reflectance. But that UVB itself is not actually visible to birds. Is there data on UVA radiation available? If not, can the authors at least elaborate on an expected association between high UVB and high UVA radiation – I suspect there must be? But I do think it would strengthen the paper if this discrepancy in the analysis was elaborated on further.

In conclusion this is a fascinating paper, and I would like to congratulate the authors on this pioneering work. I eagerly look forward to seeing more work and discoveries from this group.

Regards,
James Dale, Massey University

Reviewer #3:
Remarks to the Author:

In this manuscript the authors use an impressive and unique dataset comprising full spectrum images (UV-VIS) of over 4500 species of passerine birds to address two questions:

- (1) how is UV reflectance distributed across this sample of birds and what are its potential evolutionary drivers/correlates, and
- (2) what is the best method to separate specimen images from the background so that measurements refer only to the plumage. While I congratulate the authors on amassing such a dataset, and I have no doubt that it can be used to answer many fundamental questions on colour evolution which would be of broad interest to the readership of Nature Communications, I feel that the current manuscript falls short of this.

Disclaimer: I have no expertise in deep learning so I am unable to evaluate this part of the manuscript (which turns out to be much more prominent than what I thought).

My main concerns are as follows:

1) The manuscript feels disjointed. On the one hand it reports on the best approach to extract backgrounds from images of study skins, while on the other it analyses variation in UV reflectance and its correlates. The second feels a bit like an excuse to publish the former, and fitting both into the same manuscript means that both messages get lost. I would suggest to publish methodological aspects of the manuscript separate from biological questions.

2) As indicated above I am not familiar with the methods used to segment the images, but as a potential end user there are some aspects that I miss. Specifically, quantitative discussion on whether certain kinds of colours are more likely to cause problems and whether changes in the background used can help with segmentation. I could imagine that using a very different background from the dominant colours of the specimen would make segmentation easier.

3) Key references that have examined UV prevalence in birds and its correlates are ignored. (Eaton & Lanyon, 2003; Gomez & Théry, 2004; Mullen & Pohland, 2008; Burns & Shultz, 2012). Some of these already mention patterns described in the manuscript (phylogenetic effects, widespread UV, potential association between UV-sensitivity and UV reflectance, effects of habitat on UV). The authors should incorporate these references (and others mentioned below) into the manuscript Introduction and Discussion.

4) The predictions concerning the potential biotic and abiotic drivers of UV coloration in birds are too superficial and, in some cases, seem wrong (sensory drive) or overly simplistic (effect of solar UV radiation). This part of the paper feels very incomplete I guess due to space issues (a lot more space is used to describe methodological aspects). Currently, some conclusions do not seem to be borne by the data (that variation in UV reflectance follows predictions from Endler's sensory drive hypothesis) and the implications of others remain unclear (relevance of dorsal/ventral differences in UV or correlations with UVB radiation that is invisible to birds).

5) Some of the methods used to classify colours as having UV or not rely on largely arbitrary thresholds. I think that it would be good to assess whether results are robust to variations in these thresholds.

Specific comments (Ln refers to line number on the manuscript). Note that some of these comments may overlap with the main points outlined above.

Ln 17 I am not sure that you can call UV coloration a mechanism, revise writing. (also in Ln 50 elsewhere).

Ln 52-54 not quite true see for example: (Eaton & Lanyon, 2003; Mullen & Pohland, 2008; Burns & Shultz, 2012)

Ln 54-58 These predictions are very superficial and not sufficiently justified. You need to elaborate on each one of them, providing the existing evidence for or against (and as indicated above some evidence exists). Specifically:

- Higher UV-reflectance in species with UVS visual sensitivities: refer to (Mullen & Pohland, 2008; Odeen, Pruett-Jones, Driskell, Armenta, & Hastad, 2012; Bleiweiss, 2014; Lind & Delhey, 2015).
- Higher UV reflectance in species that inhabit regions rich in UV irradiance. While some level of UV irradiance is necessary I have not seen any previous suggestion that this is a limiting factor in diurnal birds unless in the context of vegetation structure (see below). I would welcome further elaboration and justification here.
- Higher UV reflectance in more vegetated environments. This prediction is confusing since forested environments can also be extremely UV poor (i.e. forest floor and understorey). The difference in UV reflectance between canopy and understorey species is already described (Gomez & Théry, 2004, 2007). Hence, a large proportion of forest species will inhabit some of the most UV-poor light environments. On the other hand most species of open environments will have abundant UV irradiance available, and open environments seem to favour colours richer in shorter wavelengths (McNaught & Owens, 2003). This prediction needs to be properly justified. Currently, it does not make much sense.

Ln 69 define deep learning

Ln 75-90 I am not familiar with these computational techniques, but this paragraph does not seem to be aimed at explaining these procedures to biologists. To the uninitiated this becomes very hard to follow.

Ln 102-114 I would have a similar comment as before. The authors should attempt to make these results intelligible to the general readership. It would be great to define terms such as IOU, precision, and recall here.

Ln 174 you mean all cones having 0.25 right? Clarify.

Ln 170-204 define here what you mean with peak and average values, and "UV colouration metric", so that the reader does not have to go to methods first (since these are at the end). I think that a figure with some graphical examples would be good so that the reader can better grasp the differences between these metrics.

Ln 217-218 refer to previous work here.

Ln 223-224 I do not see the results as supporting the sensory drive hypothesis as explained above.

Ln 227 as far as I understand birds do not see UVB radiation, so why did you use it? This may be relevant for photoprotection but not for signalling and certainly not the sensory drive hypothesis.

Ln 228 what are the implication of differences between dorsal and ventral? Why does this indicate that UV is used for sexual signalling (this would be the case if there would be a significant sex*body region interaction)? Why do different UV variables yield opposite results? Discuss.

Ln 233 some dorsal regions are more conspicuous (eg rump), others are not (back). I am not sure that you can generalise.

Ln 234 why a signal 'enhancer'? Do you mean in the sense of 'amplifier'? What evidence do you have to suggest this? Why can UV colour not be part of the quality indicators?

Ln 242-245 I think that some sensitivity analyses should be carried out here. Does the effect become stronger or weaker if you re-run this test using only species for which visual sensitivity is known?

Ln 251 I would say that the new insights are rather limited and require further justification.

Ln 256 what is a GPU?

Ln 567-577 Are results robust to changing some of these thresholds? How did you come up with them? Some sensitivity analyses seem to be called for.

Ln 591 Ericsson or Hackett backbone?

Ln 602 Why UVB and not UVA? Birds can only perceive UVA as far as I know.

Ln 620-631 It would be good to repeat analyses using only species for which visual system is known.

References cited

Bleiweiss, R. (2014). Physical Alignments Between Plumage Carotenoid Spectra and Cone Sensitivities in Ultraviolet-Sensitive (UVS) Birds (Passerida: Passeriformes). *Evolutionary Biology*, 41(3), 404–424. doi:10.1007/s11692-014-9273-8

Burns, K. J., & Shultz, A. J. (2012). Widespread cryptic dichromatism and ultraviolet reflectance in the largest radiation of Neotropical songbirds: Implications of accounting for avian vision in the study of plumage evolution. *The Auk*, 129(2), 211–221. doi:10.1525/auk.2012.11182

Eaton, M. D., & Lanyon, S. M. (2003). The ubiquity of avian ultraviolet plumage reflectance. *Proceedings. Biological Sciences / The Royal Society*, 270(1525), 1721–1726. doi:10.1098/rspb.2003.2431

Gomez, D., & Théry, M. (2004). Influence of ambient light on the evolution of colour signals: Comparative analysis of a Neotropical rainforest bird community. *Ecology Letters*, 7(4), 279–284. doi:10.1111/j.1461-0248.2004.00584.x

Gomez, D., & Théry, M. (2007). Simultaneous Crypsis and Conspicuousness in Color Patterns: Comparative Analysis of a Neotropical Rainforest Bird Community. *The American Naturalist*, 169(s1), S42–S61. doi:10.1086/510138

Lind, O., & Delhey, K. (2015). Visual modelling suggests a weak relationship between the evolution of ultraviolet vision and plumage colouration in birds. *Journal of Evolutionary Biology*, 28, 715–722. doi:10.1111/jeb.12595

McNaught, M. K., & Owens, I. P. F. (2003). Interspecific variation in plumage coloration among birds: species recognition or light environment? *Journal of Evolutionary Biology*, 15, 505–514.

Mullen, P., & Pohland, G. (2008). Studies on UV reflection in feathers of some 1000 bird species: Are UV peaks in feathers correlated with violet-sensitive and ultraviolet-sensitive cones? *Ibis*, 150(1), 59–68. doi:10.1111/j.1474-919X.2007.00736.x

Odeen, a., Pruett-Jones, S., Driskell, a. C., Armenta, J. K., & Hastad, O. (2012). Multiple shifts between violet and ultraviolet vision in a family of passerine birds with associated changes in plumage coloration. *Proceedings of the Royal Society B: Biological Sciences*, 279(1732), 1269–1276. doi:10.1098/rspb.2011.1777

REVIEWER COMMENTS

Reviewer #1 (Remarks to the Author):

This is a well-written and clear manuscript that describes an application of segmentation to understand the evolution of UV plumage. There are some major areas that need to be addressed, however.

It was unclear to me whether the dataset images are available (both originals and “expert-labeled”) - they must be made public so that other groups can use them to either reproduce the work or use them with other ways of performing segmentation.

- We thank the reviewer for raising this. We now make all original and expert-labelled (‘ground truth’) images used to train our model, along with the resulting machine-predicted coordinates used to test its performance, available at <https://doi.org/10.15131/shef.data.19221699>.

I think it’s unfortunate that the method doesn’t seem particularly robust to images taken differently and to rotated specimens. It seems like rotation of specimens could be corrected for in an image processing step in the workflow to mitigate this problem. The authors do spend some time exploring image augmentation, but it is still unclear to me why the results would be worse when image augmentation is implemented. I think the authors should spend some more time explaining why this was the case. Did they try all the potential augmentations offered by DeepLab?

- The reviewer mentions two separate elements of our analysis here. In relation to the first point concerning model performance on rotated images, we found that accuracy was lower compared to performance on our highly standardised dataset as the reviewer suggests, but only very marginally so (1-2%). This led us to conclude that applying our model on low-quality images did not generate excessively inaccurate predictions and was robust when used on less consistent datasets (lines 382-389). We have edited the text in this section to avoid the risk of confusion. Nonetheless, the reviewer makes a valuable suggestion regarding additional workflow steps to increase the consistence of training images, which we have incorporated into our manuscript (line 385-387).

The reviewer also discusses our results in relation to image augmentation, which is a separate analysis where we tested the performance of models trained on a larger image set including manipulated versions of existing images to boost training set size. In this regard, we did discuss possible reasons why results were (very marginally worse) when based on models trained using augmented datasets (see lines 370-378), but to address the reviewer’s point explicitly we have added additional discussion in the Supplementary Information where the image augmentation results are presented (SI lines 56-60)

Finally, we did not try all possible image augmentations, of which there are many different types, as this was outside the scope of our analysis. However, given the relatively high performance of models trained on our unaugmented dataset, and the relatively minor changes in performance based on augmented datasets, we do not believe that other augmentation approaches are likely to result in drastically improved performance.

-The authors use DeepLabv3+, but do not discuss other potential segmentation frameworks in detail. I think it is important that they describe why they chose DeepLab over other methods, what

advantages and disadvantages it has over other methods. Because many researchers are just starting to get into deep learning without deep knowledge of the frameworks and methods, more background on the options and pros and cons of different approaches is necessary.

- We have now added discussion of different algorithms and justified why we used DeepLabv3+ (Line 326-336). When we conducted the deep learning assessment (ca. 2020) DeepLabv3+ was the best performing deep learning approach when applied to standard benchmarks (the PASCAL 2012 data set). One recent approach does show some marginal improvements on benchmark datasets but DeepLabv3+ is widely used and provides a benchmark for future applications to the plumage data and other natural history data sets.

-I think there should be more discussion of prior research on UV coloration in birds, even if in much smaller scale than the work presented here. There is not much discussion about how this maps onto the phylogeny, whether there are surprises in particular clades, whether there are other common traits that appear to be correlated. While the segmentation work itself is interesting, the connection to the evolutionary story is essential and somewhat lacking here.

- In line with comments made by other reviewers, we have greatly expanded both the introduction and discussion associated with the UV colouration component of the manuscript. See our responses to specific comments below and the new text added in lines 45-81 and 260-304.

Small comments:

-In the text, DeepLab is referred to in a few ways, and because there are multiple versions of the software, this should be clarified, whether it is DeepLab, DeepLabv3, etc.

- We have clarified the versions of DeepLab models in the paper.

-In line 71, “and that allows objective measurements of colour.” Deep learning models are not objective - this should be made clear.

- We have removed “objective” in the sentence.

Reviewer #2 (Remarks to the Author):

This is an exciting paper that represents a significant advance in the field. I really like the deep learning approach to extracting data from numerous images and the comparison with other approaches is insightful and provides strong justification for the use of the deep learning data over classical segmentation methods for the biological analysis. The authors conclude that UV-signalling in avian plumage is widespread, phylogenetically conserved, sexually dimorphic, dorsally predominant and associated with the light environment (UVB and forest dependency). It is well written, and the analysis is sophisticated and state-of-the-art.

I do not have any major concerns with this paper; however I make a few comments below for the authors to consider in a revision.

1. Lines 48 to 49. Worth mentioning here the specific wavelengths that the UVS cone is sensitive to. Although it is widely known in the field that human visible light is ~ 400 to 700 nm, and for

some birds it can range from ~ 320 to 700, it might be worth describing this around here in more detail for a more general audience.

- We thank the reviewer for this suggestion and have added further detail on avian UV sensitivity (lines 47-51).

2. Lines 54-58. Predictions. Include predictions about body regions and sexual dimorphism?

- Again, we thank the reviewer for this suggestion, which was also highlighted by reviewer #3. In response to these comments, we now include an expanded introductory paragraph outlining our predictions relating to all predictor variables (see lines 59-81).

3. Lines 97 – changes “100’s of thousands” to the actual number.

- We apologise for the imprecise and inaccurate reporting and have corrected this to 146,652 (24,442 specimens with photos from three angles and each in the UV and human-vis spectrum, line 127).

4. Lines 178-180 and elsewhere in the paper. The authors mention a few times how UV reflectance “may” co-occur with reflectance at other wavelengths and I was curious about how birds with strong UV signals may appear to us? Is UV reflectance associated with violet and blue birds for instance. A figure with Average u, Peak u, and UV colouration vs reflectance from the wavelengths associated with the S, M, and L cones should be possible with their data set. This could provide insight into the private channel hypothesis mentioned in the discussion (see below). Additional mention of this is in Fig 3, where the authors state “Purple dots on the end of bars (‘UV colouration’) indicate the occurrence of detectable peaks in UV reflectance possibly occurring in combination with other colours (e.g. UV-yellow).”. I might be missing something here, but I do not understand why this can not be tested more specifically.

- We agree with the reviewer that testing the relationship between UV reflectance and reflectance at other wavelengths would be interesting. However, after careful consideration we hesitate from going down this route because we feel that analysing correlations between trait axes (i.e. different colour cones) and how they relate to the private channel hypothesis is a complex issue that would be better addressed with a focused set of analyses. We have therefore toned-down the sections relating to this idea in our paper.

5. Lines 218-222 and first part of next paragraph. Following up from the last comment, it is noteworthy that the examples of clades with notable extent of UV reflectance (tanagers, corvids, and thrushes) are highly colourful overall in the Vis-channels, and the ones with low UV reflectance (larks, and woodcreeper/ovenbirds) are generally pretty drab and brownish. In the next paragraph the authors appear to argue that the higher extent of UV on the dorsal regions are consistent with the private channel hypothesis (lines 235-238). But I think an important part of the story here has not been explored in enough detail – in that UV signalling may be associated with high conspicuousness in the Vis-light channels. If that is the case, then it would be unsupportive of the private channel hypothesis because the UV-signals will also come along with highly visible colours that would still be detectable to eavesdropping receivers like predators. The authors can test this directly: Is UV a special secret channel, or is it simply an additional way for birds to be colourful, (that happens to be invisible to humans but that we can sort of still infer in the RGB channels)?

- As mentioned above, we feel that these are very interesting questions raised by the reviewer but are outside the scope of the current analysis and would be better off followed up on with a focused set of analyses. Consequently, we have revised the text to remove direct references to the private channel hypothesis and instead provide a much more focused discussion of our results, particularly in relation to dorso-ventral differences in UV reflectance (lines 284-304) that were not adequately discussed previously (see comment from Reviewer #3 below).

Line 252 – I got tripped up by this sentence – maybe insert comma between “lie” and “rely

- Comma added.

Lines 338 and elsewhere. The plumage UV reflectance is measured across 320-380nm by the Baader U-Venus-Filtre, and these wavelengths will be visible to birds with the UVS cone and correspond to UVA light (~320-400nm). But one of the key environmental predictors used is annual mean UVB radiation (~280-320nm) – and it is found to be associated with UVA reflectance. But that UVB itself is not actually visible to birds. Is there data on UVA radiation available? If not, can the authors at least elaborate on an expected association between high UVB and high UVA radiation – I suspect there must be? But I do think it would strengthen the paper if this discrepancy in the analysis was elaborated on further.

- We agree that our use of UVB was not explained. It is indeed the case that UVB and UVA are closely correlated and our choice to use UVB data was pragmatic based on the availability of readily usable data. We have added the following to the methods to clarify this point (lines 692-694): “We used global data on UVB radiation as a proxy for UVA. UVA and UVB have highly similar global distributions and we use UVB due to the availability of data designed for macroecological studies.”

In conclusion this is a fascinating paper, and I would like to congratulate the authors on this pioneering work. I eagerly look forward to seeing more work and discoveries from this group.

- We thank the reviewer for their positive appraisal of our manuscript!

Reviewer #3 (Remarks to the Author):

In this manuscript the authors use an impressive and unique dataset comprising full spectrum images (UV-VIS) of over 4500 species of passerine birds to address two questions:

(1) how is UV reflectance distributed across this sample of birds and what are its potential evolutionary drivers/correlates,

(2) what is the best method to separate specimen images from the background so that measurements refer only to the plumage. While I congratulate the authors on amassing such a dataset, and I have no doubt that it can be used to answer many fundamental questions on colour evolution which would be of broad interest to the readership of Nature Communications, I feel that the current manuscript falls short of this.

Disclaimer: I have no expertise in deep learning so I am unable to evaluate this part of the manuscript (which turns out to be much more prominent than what I thought).

My main concerns are as follows:

1) The manuscript feels disjointed. On the one hand it reports on the best approach to extract backgrounds from images of study skins, while on the other it analyses variation in UV reflectance and its correlates. The second feels a bit like an excuse to publish the former, and fitting both into the same manuscript means that both messages get lost. I would suggest to publish methodological aspects of the manuscript separate from biological questions.

- We acknowledge this view and it is something that we spent a great deal of time discussing before writing the paper. The reason that we put the methodological and biological components together is that we want to illustrate the value of the methods by providing a real-world example (application) of using this dataset (whole body plumage colour and UV).

2) As indicated above I am not familiar with the methods used to segment the images, but as a potential end user there are some aspects that I miss. Specifically, quantitative discussion on whether certain kinds of colours are more likely to cause problems and whether changes in the background used can help with segmentation. I could imagine that using a very different background from the dominant colours of the specimen would make segmentation easier.

- We thank the reviewer for this valuable suggestion. We have now added a new analysis to explore whether the level of contrast between specimens and the background affects deep learning performance. We found that model performance was generally high across all levels of specimen-background contrast but marginally declined with increasing contrast values (see new Supplementary Fig. 7). The slight decline in model performance with increasing contrast is opposite to the expectation that high-contrast specimens should be easier to segment than low-contrast specimens but can potentially be explained by lower sample sizes at higher contrasts resulting in more limited opportunities for our model to learn to accurately segment high-contrast birds. Relative to classic segmentation methods, however, the deep learning approach is far more robust to specimen-background contrast issues, particularly at low values (Supplementary Fig. 7). We report these new results briefly in the main text (lines 183-186) and provide detailed methods information and results in the supplementary information (SI lines 109-145).

3) Key references that have examined UV prevalence in birds and its correlates are ignored. (Eaton & Lanyon, 2003; Gomez & Théry, 2004; Mullen & Pohland, 2008; Burns & Shultz, 2012). Some of these already mention patterns described in the manuscript (phylogenetic effects, widespread UV, potential association between UV-sensitivity and UV reflectance, effects of habitat on UV). The authors should incorporate these references (and others mentioned below) into the manuscript Introduction and Discussion.

- We thank the reviewer for pointing out these important oversights. As noted in the specific responses below, we have re-written parts of the introduction and discussion to reflect the literature and state of the art more accurately (see specific responses below), citing the papers mentioned by the reviewer.

4) The predictions concerning the potential biotic and abiotic drivers of UV coloration in birds are too superficial and, in some cases, seem wrong (sensory drive) or overly simplistic (effect of solar UV radiation). This part of the paper feels very incomplete I guess due to space issues (a lot more space is used to describe methodological aspects). Currently, some conclusions do not seem to be borne by the data (that variation in UV reflectance follows predictions from Endler's sensory drive hypothesis) and the implications of others remain unclear (relevance of dorsal/ventral differences in UV or correlations with UVB radiation that is invisible to birds).

- To address this comment we have now extensively revised and expanded the sections of our manuscript that introduce our predictions (lines 59-81) and discuss our findings in relation to these predictions (lines 269-315). In particular, we have taken care to explain how our predictions relate to Endler's sensory drive hypothesis and to discuss instances where our results do and do not follow these predictions.

5) Some of the methods used to classify colours as having UV or not rely on largely arbitrary thresholds. I think that it would be good to assess whether results are robust to variations in these thresholds.

- The reviewer raises an important methodological point here. In response we have run new analyses for the two variables in question (peak u and UV+ colouration) using alternative thresholds (see lines 663-667 for details). Encouragingly, we found that our results were consistent irrespective of the threshold used. These new results have been added to Supplementary Tables 3 and 4 and are mentioned in the main text (line 246-247).

Specific comments (Ln refers to line number on the manuscript). Note that some of these comments may overlap with the main points outlined above.

Ln 17 I am not sure that you can call UV coloration a mechanism, revise writing. (also in Ln 50 elsewhere).

- Agreed and we have rephrased to refer to "form of signalling" rather than "signalling mechanism".

Ln 52-54 not quite true see for example: (Eaton & Lanyon, 2003; Mullen & Pohland, 2008; Burns & Shultz, 2012)

- We thank the reviewer for pointing this out. We have re-written this sentence to more accurately reflect the literature and cited the suggested literature (lines 54-58).

Ln 54-58 These predictions are very superficial and not sufficiently justified. You need to elaborate on each one of them, providing the existing evidence for or against (and as indicated above some evidence exists). Specifically:

- Higher UV-reflectance in species with UVS visual sensitivities: refer to (Mullen & Pohland, 2008; Odeen, Pruett-Jones, Driskell, Armenta, & Hastad, 2012; Bleiweiss, 2014; Lind & Delhey, 2015).
- Higher UV reflectance in species that inhabit regions rich in UV irradiance. While some level of UV irradiance is necessary I have not seen any previous suggestion that this is a limiting factor in diurnal birds unless in the context of vegetation structure (see below). I would welcome further elaboration and justification here.
- Higher UV reflectance in more vegetated environments. This prediction is confusing since forested environments can also be extremely UV poor (i.e. forest floor and understorey). The difference in UV reflectance between canopy and understorey species is already described (Gomez & Théry, 2004, 2007). Hence, a large proportion of forest species will inhabit some of the most UV-poor light environments. On the other hand most species of open environments will have abundant UV irradiance available, and open environments seem to favour colours richer in shorter wavelengths (McNaught & Owens, 2003). This prediction needs to be properly justified. Currently, it does not make much sense.

- As mentioned above, we now include a greatly expanded introductory paragraph outlining our reasoning and predictions relating to all of our predictor variables, citing many of the papers mentioned by the reviewer (see lines 59-81). Also, in relation to the reviewer's third point regarding the expected difference between canopy and understorey species, we have added a new variable capturing variation in typical foraging height between species (see lines 749-752 for methods details). Testing relationships between this variable and our UV metrics reveals that upper storey and canopy foraging species do indeed have higher levels of UV reflectance than understorey species, in line with sensory drive theory and the reviewer's suggestion. These results are described in line 239 and have been integrated into our reworked discussion.

Ln 69 define deep learning

- We have now added a definition of deep learning (see lines 92-93).

Ln 75-90 I am not familiar with these computational techniques, but this paragraph does not seem to be aimed at explaining these procedures to biologists. To the uninitiated this becomes very hard to follow.

- We agree that some aspects of this paragraph may be hard to follow. To avoid further expanding the main manuscript with more detail on computational techniques we have added a table that summarises the key methods to the supplementary materials (Supplementary Table 1).

Ln 102-114 I would have a similar comment as before. The authors should attempt to make these results intelligible to the general readership. It would be great to define terms such as IOU, precision, and recall here.

- We agree that some of the terminology was not adequately defined. We now give brief definitions of IOU, precision, and recall and direct reader to the methods section for detailed definitions (lines 133-136).

Ln 174 you mean all cones having 0.25 right? Clarify.

- Correct. We have clarified the text to read: "where cone-catch values of 0.25 for all cones would be considered the achromatic null." (lines 215-216).

Ln 170-204 define here what you mean with peak and average values, and "UV colouration metric", so that the reader does not have to go to methods first (since these are at the end). I think that a figure with some graphical examples would be good so that the reader can better grasp the differences between these metrics.

- We thank the reviewer for these suggestions. We now define each of the three metrics here (see lines 208-213), as well as pointing readers to the Methods section for more details.

Ln 217-218 refer to previous work here.

- We have now added references (Burns and Schultz 2012, Eaton and Lanyon 2003, Mullen and Pohland 2008) and acknowledged previous advances (lines 260-268).

Ln 223-224 I do not see the results as supporting the sensory drive hypothesis as explained above.

- We now outline in detail in the introduction how our predictions (and consequently our results) relate to the sensory drive hypothesis (see lines 59-81).

Ln 227 as far as I understand birds do not see UVB radiation, so why did you use it? This may be relevant for photoprotection but not for signalling and certainly not the sensory drive hypothesis.

- Reviewer 2 raised a similar point. We repeat our response: "We agree that our use of UVB was not explained. It is indeed the case that UVB and UVA are closely correlated and our choice to use UVB data was pragmatic based on the availability of readily usable data. We have added the following to the methods to clarify this point (lines 692-694): "We used global data on UVB radiation as a proxy for UVA. UVA and UVB have highly similar global distributions and we use UVB due to the availability of data designed for macroecological studies."

Ln 228 what are the implication of differences between dorsal and ventral? Why does this indicate that UV is used for sexual signalling (this would be the case if there would be a significant sex*body region interaction)? Why do different UV variables yield opposite results? Discuss.

- We agree that these results were not adequately discussed in the previous version of our manuscript. We have therefore extensively revised this section of our manuscript to discuss in detail both the implication of the dorso-ventral differences and the observation that different UV variables yield different results (lines 284-304).

Ln 233 some dorsal regions are more conspicuous (eg rump), others are not (back). I am not sure that you can generalise.

- As part of the changes mentioned above (see previous comment), this section has now been extensively revised (lines 284-304).

Ln 234 why a signal 'enhancer'? Do you mean in the sense of 'amplifier'? What evidence do you have to suggest this? Why can UV colour not be part of the quality indicators?

- Again, this section has now been extensively revised in response to comments from multiple reviewers (lines 284-304).

Ln 242-245 I think that some sensitivity analyses should be carried out here. Does the effect become stronger or weaker if you re-run this test using only species for which visual sensitivity is known?

- This is an interesting suggestion but there are so few passerine species for which visual sensitivities are explicitly known (and that overlap with our dataset) that in our view this would provide limited insight. However, the sparseness of visual system data across passerines is an important consideration so we have added mention of this to our discussion of this result (lines 312-315).

Ln 251 I would say that the new insights are rather limited and require further justification.

- We hope that the changes we have made in response to the specific points raised above have addressed this comment.

Ln 256 what is a GPU?

-We now provide a definition of GPU (graphics processing unit) on line 323.

Ln 567-577 Are results robust to changing some of these thresholds? How did you come up with them? Some sensitivity analyses seem to be called for.

- As described above, we now include sensitivity analyses using alternative thresholds and find that our results are robust to the precise thresholds used (see Supplementary Tables 3 and 4).

Ln 591 Ericsson or Hackett backbone?

- We used the Hackett backbone – this has been added (line 680).

Ln 602 Why UVB and not UVA? Birds can only perceive UVA as far as I know.

- We refer to our earlier responses and addition of text to justify this in the methods section.

Ln 620-631 It would be good to repeat analyses using only species for which visual system is known.

- As mentioned above, while this is an interesting suggestion, in practice the sparseness of available visual system data for passerine species means that this analysis would provide limited insight in our view. However, as noted we now explicitly mention the sparseness issue in our discussion (lines 312-315).

\Reviewers' Comments:

Reviewer #1:

Remarks to the Author:

The authors have adequately addressed reviewer comments.

Reviewer #2:

Remarks to the Author:

I would like to thank the authors for addressing all my comments on the manuscript. I also reviewed the responses made to the other two reviewers and these also seemed fine. I will certainly be looking forward to seeing further work on how UV signaling co-varies with other colour channels!

Reviewer #3:

Remarks to the Author:

The authors have addressed several of my concerns in this new version of the manuscript. I do still feel that combining methodological aspects on image segmentation with analyses trying to explain variation in UV coloration in birds leads to a hybrid paper that is neither here nor there. The risk is that both messages get lost. That is however, only my opinion.

More worryingly I still feel that the biologically-relevant part of the paper (trying to explain UV coloration across species) receives a very superficial treatment. In my previous review I suggested that the authors incorporate and discuss previous attempts at dealing with this question. While the references have been cited I was hoping for a bit more in detail discussion on how the results presented in this manuscript fit or not with previous knowledge.

In addition, I was puzzled by the prediction that we should expect a positive correlation between plumage UV reflectance and global variation in UVB levels and I asked for further clarification behind this. Unfortunately, I still think that this part has not been fully thought through. The authors argue that we should expect a correlation between plumage UV reflectance and (mean annual?) UVB radiation within each species range. I do not think that this test is properly justified and I suspect that the correlation, rather than being causal, may be driven by other correlated variables. Specifically, in this regard I point to the following issues:

1) They justify testing for this effect based on the fact that correlations between UVB and pigmentation in birds have been found (Nicolai et al. 2020). Citing this reference is misleading as it concerns skin pigmentation in terms of the photoprotection hypothesis (increased melanin deposition helps protecting from UV-induced damage), and hence it has nothing to do with the sensory drive hypothesis being tested here. If the authors refer to this study they need to clarify this.

2) While they correctly indicate (In 64-65) that "ambient light with proportionally high levels of UV should favour the use of UV signals" based on Endler's sensory drive theory, they seemingly use total absolute levels of UVB radiation as a correlate (from Beckmann et al. 2014). While their use of UVB rather than the visually relevant UVA may be justified if both are strongly positively correlated [however reading (Jablonski & Chaplin, 2010), it seems that there are also quite a few differences in the spatial and seasonal patterns of variation in UVA and UVB and I could not find a quantitative assessment of this correlation], it should be relative UV that matters. As a matter of fact, Annual mean radiation correlates strongly with UVB levels ($r=0.82$, Beckmann et al. 2014) on a spatial scale, so computing relative levels of UVB may actually reveal a different pattern than the one reported in the manuscript.

3) UVB levels also correlate strongly ($r=0.84$) with annual mean temperature (Beckmann et al. 2014) and most likely latitude as well. Climatic/latitudinal patterns in coloration have been hypothesised/found/dismissed in different studies involving a variety of different selection forces (Gloger, 1833; Bailey, 1978; Dalrymple et al., 2015, 2018; Galván et al., 2018; Delhey et al., 2019). Simply assuming that a positive correlation between total UVB and relative plumage UV reflectance is supportive of the sensory drive hypothesis (Ln 273-274) is misleading and too superficial.

Taken together these issues indicate to me that the conclusion (a positive correlation between UVB and plumage UV supporting Endler's sensory drive hypothesis) is not warranted. Further tests/explanations are required.

Specific suggestions (Ln=line number)

Ln 133-136 I still think that you could further clarify what each of these measurements mean by bringing in more detail from the methods section. Without other context it is unclear to the uninitiated reader what 'sensitivity' or 'positive predictive value' mean.

Ln 235-248 Please add estimates of marginal (fixed effects) and conditional (fixed + random effects) R² values to the results.

Ln 299 'front-facing regions dominated by carotenoid based pigments' I am not aware of any supporting data for this statement, please cite the relevant paper. Note as well that it should be either "carotenoid based colours" or "carotenoid pigments".

Ln 299-304 this is anecdotal and does not constitute a good explanation. One could also come up with alternative examples.

References

- Bailey, F. 1978. Latitudinal gradients in colors and patterns of passerine birds. *Condor* 80: 372–381.
- Dalrymple, R.L., Flores-Moreno, H., Kemp, D.J., White, T.E., Laffan, S.W., Hemmings, F.A., et al. 2018. Abiotic and biotic predictors of macroecological patterns in bird and butterfly coloration. *Ecol. Monogr.* 88: 204–224.
- Dalrymple, R.L., Kemp, D.J., Flores-Moreno, H., Laffan, S.W., White, T.E., Hemmings, F.A., et al. 2015. Birds, butterflies and flowers in the tropics are not more colourful than those at higher latitudes. *Glob. Ecol. Biogeogr.* 24: 1424–1432.
- Delhey, K., Dale, J., Valcu, M. & Kempenaers, B. 2019. Reconciling ecogeographical rules: rainfall and temperature predict global colour variation in the largest bird radiation. *Ecol. Lett.* 22: 726–736.
- Galván, I., Rodríguez-Martínez, S. & Carrascal, L.M. 2018. Dark pigmentation limits thermal niche position in birds. *Funct. Ecol.* 32: 1531–1540.
- Gloger, C.W.L. 1833. *Das Abändern der Vögel durch Einfluss des Klimas.* August Schulz., Breslau.
- Jablonski, N.G. & Chaplin, G. 2010. Human skin pigmentation as an adaptation to UV radiation. 107.

REVIEWER COMMENTS

Reviewer #1 (Remarks to the Author):

The authors have adequately addressed reviewer comments.

> We thank the reviewer for their constructive feedback.

Reviewer #2 (Remarks to the Author):

I would like to thank the authors for addressing all my comments on the manuscript. I also reviewed the responses made to the other two reviewers and these also seemed fine. I will certainly be looking forward to seeing further work on how UV signaling co-varies with other colour channels!

> We are pleased that the reviewer is happy with our revisions and again would like to thank them for their extremely useful comments.

Reviewer #3 (Remarks to the Author):

The authors have addressed several of my concerns in this new version of the manuscript. I do still feel that combining methodological aspects on image segmentation with analyses trying to explain variation in UV coloration in birds leads to a hybrid paper that is neither here nor there. The risk is that both messages get lost. That is however, only my opinion.

> We are pleased that our revisions have addressed several of the reviewer's concerns. We again acknowledge the reviewer's perspective regarding the structure of our manuscript but feel that illustrating the value of our methods by providing a real-world application using this dataset as part of the same manuscript is valuable.

More worryingly I still feel that the biologically-relevant part of the paper (trying to explain UV coloration across species) receives a very superficial treatment. In my previous review I suggested that the authors incorporate and discuss previous attempts at dealing with this question. While the references have been cited I was hoping for a bit more in detail discussion on how the results presented in this manuscript fit or not with previous knowledge.

> We recognise that our study is by no means a definitive explanation of the correlates and causes of UV colouration across species. We believe that the addition of the temperature, solar radiation and relative UVB analyses, suggested by the reviewer, have added depth and insight to our results. Furthermore, we have added further discussion of our results, particularly in relation to the new variables we include (temperature and solar radiation; see lines 308-322). There is limited space to expand further in the text and we hope that our studies prompts future work that addresses specific emergent hypotheses and predictions.

In addition, I was puzzled by the prediction that we should expect a positive correlation between plumage UV reflectance and global variation in UVB levels and I asked for further clarification behind this. Unfortunately, I still think that this part has not been fully thought through. The authors argue that we should expect a correlation between plumage UV reflectance and (mean annual?) UVB radiation within each species range. I do not think that this test is properly justified and I suspect that the correlation, rather than being causal, may be driven by other correlated variables. Specifically, in this regard I point to the following issues:

1) They justify testing for this effect based on the fact that correlations between UVB and pigmentation in birds have been found (Nicolai et al. 2020). Citing this reference is misleading as it concerns skin pigmentation in terms of the photoprotection hypothesis (increased melanin deposition helps protecting from UV-induced damage), and hence it has nothing to do with the sensory drive hypothesis being tested here. If the authors refer to this study they need to clarify this.

> We appreciate the reviewer's comment on our use of the Nicolai et al. 2020 study (reference #8 in our manuscript). It was not our intention to use it to imply a link between UVB and sensory drive, rather we use it to highlight that UV and solar radiation has been shown to exert selection pressure on colouration. We recognise that our phrasing was ambiguous and have clarified the text to emphasise that this study relates specifically to the photoprotection hypothesis (line 55 and 74 in revised version).

2) While they correctly indicate (ln 64-65) that "ambient light with proportionally high levels of UV should favour the use of UV signals" based on Endler's sensory drive theory, they seemingly use total absolute levels of UVB radiation as a correlate (from Beckmann et al. 2014). While their use of UVB rather than the visually relevant UVA may be justified if both are strongly positively correlated [however reading (Jablonski & Chaplin, 2010), it seems that there are also quite a few differences in the spatial and seasonal patterns of variation in UVA and UVB and I could not find a quantitative assessment of this correlation], it should be relative UV that matters. As a matter of fact, Annual mean radiation correlates strongly with UVB levels ($r=0.82$, Beckmann et al. 2014) on a spatial scale, so computing relative levels of UVB may actually reveal a different pattern than the one reported in the manuscript.

> Please see our response to the next comment, where we address this comment and the next comment simultaneously.

3) UVB levels also correlate strongly ($r=0.84$) with annual mean temperature (Beckmann et al. 2014) and most likely latitude as well. Climatic/latitudinal patterns in coloration have been hypothesised/found/dismissed in different studies involving a variety of different selection forces (Gloger, 1833; Bailey, 1978; Dalrymple et al., 2015, 2018; Galván et al., 2018; Delhey et al., 2019). Simply assuming that a positive correlation between total UVB and relative plumage UV reflectance is supportive of the sensory drive hypothesis (ln 273-274) is misleading and too superficial.

> The reviewer raises several important points here, in that previously we investigated the effect of variation in absolute UVB levels, not relative UVB levels as highlighted by theory, and that an effect of UVB on colouration could emerge due to correlated effects of unstudied variables (specifically temperature). Therefore we have followed the reviewers' suggestion and included additional variables in our multi-predictor models capturing variation in both total annual solar radiation exposure and annual mean temperature. This allows us to control for correlated variation in total solar radiation and temperature across species and therefor specifically test the effect of relative UVB on colouration. Encouragingly, our results remain broadly consistent with respect to the positive effect of (relative) UVB, with the exception that the formerly weak correlation between UVB and 'UV+ colouration' is no longer statistically significant. Therefore overall we stand by our original conclusion that relative levels of incident UVB play an important role in promoting the evolution of 'pure' UV colouration at least, which we view as being in line

with the predictions of Endler's sensory drive theory. We have updated the text in the relevant places to reflect our new results (see lines 257-271).

Taken together these issues indicate to me that the conclusion (a positive correlation between UVB and plumage UV supporting Endler's sensory drive hypothesis) is not warranted. Further tests/explanations are required.

> We hope that our new analyses and clarifications help to alleviate the reviewer's concerns. In any case, we thank the reviewer for their thoughtful critique of our approach, which has certainly helped to improve the manuscript.

Specific suggestions (Ln=line number)

Ln 133-136 I still think that you could further clarify what each of these measurements mean by bringing in more detail from the methods section. Without other context it is unclear to the uninitiated reader what 'sensitivity' or 'positive predictive value' mean.

> We have further clarified the meaning of these measurements (line 143-146).

Ln 235-248 Please add estimates of marginal (fixed effects) and conditional (fixed + random effects) R^2 values to the results.

> We have added marginal and conditional R^2 for all multi-predictor models to Table S4.

Ln 299 'front-facing regions dominated by carotenoid based pigments' I am not aware of any supporting data for this statement, please cite the relevant paper. Note as well that it should be either "carotenoid based colours" or "carotenoid pigments".

> We have added a reference to support this statement (Cooney et al. 2019, *Nature Communications* 10, 1773). We have also corrected the statement to 'carotenoid-based colours' – thank you.

Ln 299-304 this is anecdotal and does not constitute a good explanation. One could also come up with alternative examples.

> We accept the reviewer's point and have therefore edited this statement to make clear that this represents an anecdotal observation that remains to be explicitly tested (lines 347-350).

References

- Bailey, F. 1978. Latitudinal gradients in colors and patterns of passerine birds. *Condor* 80: 372–381.
- Dalrymple, R.L., Flores-Moreno, H., Kemp, D.J., White, T.E., Laffan, S.W., Hemmings, F.A., et al. 2018. Abiotic and biotic predictors of macroecological patterns in bird and butterfly coloration. *Ecol. Monogr.* 88: 204–224.
- Dalrymple, R.L., Kemp, D.J., Flores-Moreno, H., Laffan, S.W., White, T.E., Hemmings, F.A., et al. 2015. Birds, butterflies and flowers in the tropics are not more colourful than those at higher latitudes. *Glob. Ecol. Biogeogr.* 24: 1424–1432.
- Delhey, K., Dale, J., Valcu, M. & Kempenaers, B. 2019. Reconciling ecogeographical rules: rainfall and temperature predict global colour variation in the largest bird radiation. *Ecol. Lett.* 22: 726–736.
- Galván, I., Rodríguez-Martínez, S. & Carrascal, L.M. 2018. Dark pigmentation limits thermal

niche position in birds. *Funct. Ecol.* 32: 1531–1540.

Gloger, C.W.L. 1833. *Das Abändern der Vögel durch Einfluss des Klimas*. August Schulz., Breslau.

Jablonski, N.G. & Chaplin, G. 2010. Human skin pigmentation as an adaptation to UV radiation. 107.

.

Reviewers' Comments:

Reviewer #3:

Remarks to the Author:

The authors have addressed my previous concerns in a satisfactory manner. I would only add the suggestion to include and briefly discuss the model R² values in the main text.

REVIEWERS' COMMENTS

Reviewer #3 (Remarks to the Author):

The authors have addressed my previous concerns in a satisfactorily manner. I would only add the suggestion to include and briefly discuss the model R2 values in the main text.

> We now include a sentence stating the range of R2 values associated with our models in the main text (line 257-260).